# Parvalbumin neurons enhance temporal coding and reduce cortical noise in complex auditory scenes

Jian Carlo Nocon [1,2,3,4], Howard J. Gritton[5,6], Nicholas M. James[1,2,3,4], Rebecca A. Mount[1,2,3,4], Zhili Qu[5,6], Xue Han [1,2,3,4] & Kamal Sen [1,2,3,4 ✉]

Cortical representations supporting many cognitive abilities emerge from underlying circuits comprised of several different cell types. However, cell type-specific contributions to rate and timing-based cortical coding are not well-understood. Here, we investigated the role of parvalbumin neurons in cortical complex scene analysis. Many complex scenes contain sensory stimuli which are highly dynamic in time and compete with stimuli at other spatial locations. Parvalbumin neurons play a fundamental role in balancing excitation and inhibition in cortex and sculpting cortical temporal dynamics; yet their specific role in encoding complex scenes via timing-based coding, and the robustness of temporal representations to spatial competition, has not been investigated. Here, we address these questions in auditory cortex of mice using a cocktail party-like paradigm, integrating electrophysiology, optogenetic manipulations, and a family of spike-distance metrics, to dissect parvalbumin neurons' contributions towards rate and timing-based coding. We find that suppressing parvalbumin neurons degrades cortical discrimination of dynamic sounds in a cocktail party-like setting via changes in rapid temporal modulations in rate and spike timing, and over a wide range of time-scales. Our findings suggest that parvalbumin neurons play a critical role in enhancing cortical temporal coding and reducing cortical noise, thereby improving representations of dynamic stimuli in complex scenes.

[1] Neurophotonics Center, Boston University, Boston 02215 MA, USA. [2] Center for Systems Neuroscience, Boston University, Boston 02215 MA, USA. [3] Hearing Research Center, Boston University, Boston 02215 MA, USA. [4] Department of Biomedical Engineering, Boston University, Boston 02215 MA, USA. [5] Department of Comparative Biosciences, University of Illinois, Urbana 61820 IL, USA. [6] Department of Bioengineering, University of Illinois, Urbana 61820 IL, USA. ✉email: kamalsen@bu.edu

The cerebral cortex is critical for perception, attention, decision-making, memory, and motor output. Understanding the cortical circuit mechanisms that underlie these functions remains a central problem in systems neuroscience. One line of investigation toward addressing this problem has been to identify the underpinnings of the cortical code; specifically, to assess whether cortical coding relies on rate or spike timing[1]. Previous studies have demonstrated both rate and spike timing-based coding in cortex[2–4]. However, a mechanistic understanding of how cortical circuits implement these codes and on what timescales is still missing. A second line of questioning towards addressing this central problem has been to utilize a combination of anatomy, physiology and optogenetics to interrogate cortical circuits and neuron types[5,6]. This concerted approach has allowed systems neuroscientists to identify key contributions of specific cell types to cortical circuits, including inhibitory neurons (e.g., parvalbumin-expressing (PV), somatostatin-expressing (SOM), and vasoactive intestinal peptide-expressing (VIP) neurons)[5]. However, the specific contributions of these diverse cell types to the cortical code remain unclear.

A potentially powerful strategy for unraveling cell-type-specific contributions to cortical coding is to investigate problems where cortical processing is likely to play a central role. An important example of such a problem is complex scene analysis, e.g., recognizing objects in a scene cluttered with multiple objects at different spatial locations. The brain displays an astonishing ability to navigate such complex scenes in everyday settings, an impressive feat yet to be matched by state-of-the-art machines. The relative contribution of specific cell types to this powerful computational ability remains unclear.

PV neurons are the most prominent group of inhibitory neurons in the cortex[7]. Previous studies have investigated the role played by PV neurons in the generation of oscillations[8] and spike synchronization[9]. PV neurons play a fundamental role in balancing excitation and inhibition[10] and determining receptive field properties in the cortex[11–13]. Optogenetic manipulation of PV neurons has provided insights into cortical responses, network dynamics, and behavior[14–19]. Specifically, a study by Moore et al.[19] revealed that optogenetic suppression of PV neurons led to a rapid rebalancing of excitation and inhibition in the cortex, with the expected increase in the activity of excitatory neurons, but a counterintuitive increase in the activity of inhibitory neurons. As elegantly dissected in the study, this occurs because the suppression of PV neurons leads to an increase in the activity of excitatory neurons, which then drives both excitatory and inhibitory neurons downstream, rapidly rebalancing cortical activity. This result illuminates a property of cortical networks consistent with theoretical models but raises another question: does the suppression of PV neurons impact cortical temporal coding? The biophysical properties of PV neurons are well-suited for rapid temporal processing[5] and therefore may be essential in the cortical temporal coding of dynamic stimuli present in complex scenes. In addition, narrow-spiking units, which are thought to be putative inhibitory neurons, have exhibited distinct temporal response patterns to stimulus envelopes compared to those of regular-spiking units[20]. This motivates several open questions: do PV neurons play a critical role in the cortical temporal coding of dynamic stimuli? Are such temporal codes robust to competing stimuli at other locations in space? Here, we address these questions in the auditory cortex using a combination of electrophysiology, optogenetic suppression of PV neurons, and a family of spike-distance metrics[21,22] to dissect specific contributions of PV neurons to the cortical code.

The auditory cortex (ACx) is well-suited to investigate these issues. It is thought to play a key role in solving the cocktail party problem[23,24], one of the most impressive examples of complex scene analysis. Here, we integrate a cocktail party-like paradigm[25] with optogenetic suppression of PV neurons to investigate the specific contribution of PV neurons to temporal coding in mouse ACx. We find that suppressing PV neurons degrades discrimination performance, specifically temporal coding, in ACx, and degrades performance over a wide range of timescales. Our results reveal that despite the rebalancing of excitation and inhibition in cortical networks observed previously, suppression of PV neurons disrupts coding throughout ACx, suggesting an important influence of PV neurons on cortical temporal coding and regulating cortical noise.

## Results

We recorded single units (SUs) and multiunits (MUs) using a multielectrode array with 4 shanks and 32 channels throughout different layers in ACx of unanesthetized PV-Arch transgenic mice (Fig. 1a–c and Supplementary Figs. 1 and 2). We used a semi-automated detection and sorting algorithm to identify 124 units from $n = 9$ animals[26,27]. Of these 124 units, 82 were identified as SUs (e.g., Fig. 1d) while the remaining 42 were identified as MUs. In the results below, we focus on SUs. Out of the SUs, 73 were identified as regular-spiking (RS) while the remaining 9 were identified as narrow-spiking (NS) based on the trough-peak interval of their mean waveforms (Supplementary Fig. 3). RS and NS units have been found to correspond to excitatory and inhibitory neurons, respectively, in ACx[12,28].

To confirm specificity of expression, immunohistochemistry quantification was performed at the conclusion of the study and revealed that ~93% of PV immunopositive cells in the auditory cortex were also Arch-GFP expressing neurons. Importantly, we also found that <1% of Arch-GFP expressing cells were immuno-negative for PV antibody (Supplementary Fig. 4). Optogenetic suppression occurred on ~50% of trials, randomly interleaved, throughout the recording sessions. Suppression was achieved using light output from a 532 nm laser that began 50 ms prior to the auditory stimulus and consisted of continuous illuminations that co-terminated with sound offset. Within a given 800-trial session, optogenetic suppression strength remained constant (2 mW, 5 mW, or 10 mW), but was varied across sessions.

Next, we confirmed that the effects of optogenetic suppression of PV neurons in ACx were consistent with previous studies (Fig. 1e and Supplementary Figs 5 and 6). Upon laser onset, we found that NS units in PV-Arch-expressing subjects showed an immediate suppression of spiking followed by an increase in activity (Supplementary Fig. 5a), while NS units within non-Arch-expressing subjects did not show a change in activity during laser onset (Supplementary Fig. 5b). We found that upon PV suppression, RS units increased their firing rate during both spontaneous and auditory evoked periods (Fig. 1e and Supplementary Fig. 6a), as expected[18,19]. Different intensities enhanced the firing rate of RS neurons in a level-dependent manner consistent with previous studies (Supplementary Fig. 6a–c). Counterintuitively, but consistent with the previous study in ref. [19], NS units also increased their firing activity (Supplementary Fig. 6d–f). As demonstrated by Moore et al. optogenetic suppression of PV neurons also produced a compensatory increase in inhibition and a rapid rebalancing of excitation and inhibition in the cortex. Thus, the effects of optogenetic suppression of PV neurons on firing rates in ACx we observed are consistent with previous studies and the rapid rebalancing of excitation and inhibition. However, the effects of PV suppression on temporal coding in ACx remain unknown. Thus, we next inquired: does PV suppression impact temporal coding in ACx?

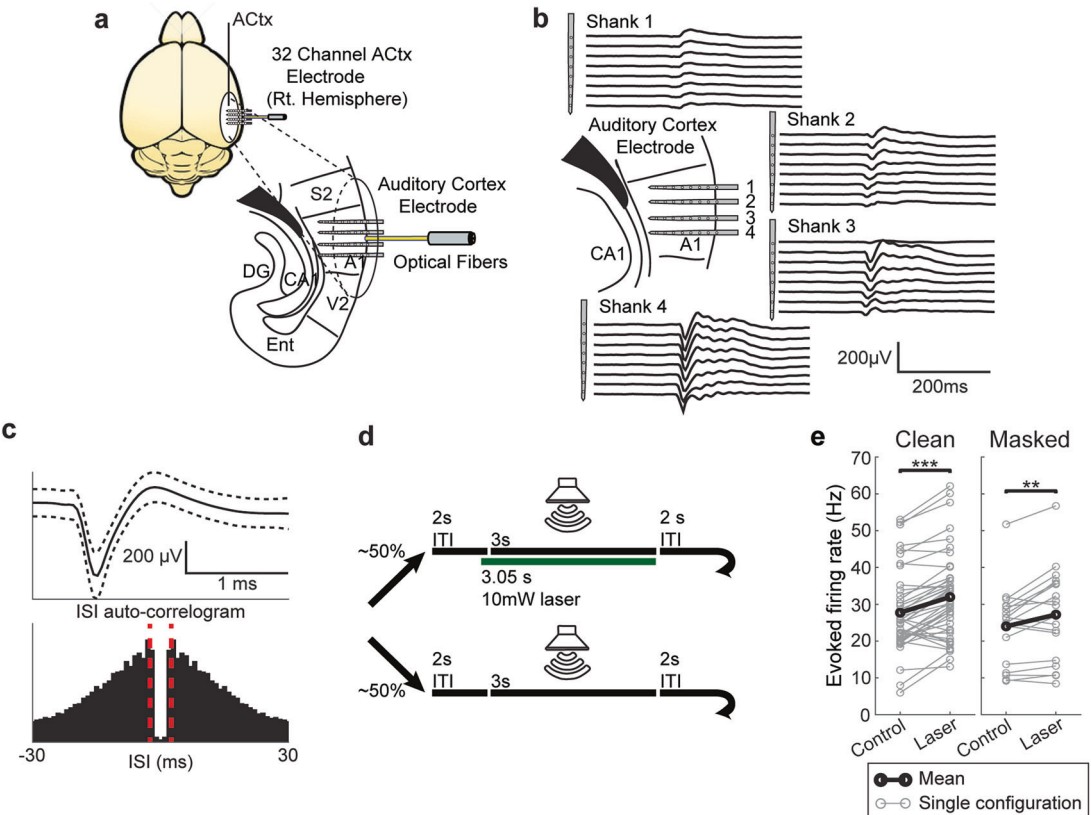

**Fig. 1 Experimental methods. a** Recording electrode location and optical fiber placement. Subjects were implanted with a 4-shank, 32-channel electrode array, and optogenetic fiber in the right hemisphere of ACx. Each shank contained 8 sites per shank with 100 μm spacing between electrode contacts. Mouse brain illustration from Pixta (https://www.pixtastock.com/illustration/67155575). **b** Representative local field potential (LFP) activity from one mouse. LFP was used to estimate current source density and the layer of the recording site within each shank (Supplementary Fig. 2). **c** Example mean single-unit waveform and inter-spike interval (ISI) auto-correlogram. Dashed lines in mean waveform represent one standard deviation above and below the mean, while scale bars are equal to 200 μV and 1 ms. Dashed red lines in the correlogram represent ISIs of ±2 ms. **d** Schematic for control and optogenetic trial presentation. During approximately 50% of all trials, a 532 nm laser would turn on 50 ms before sound stimulus onset and turn off coincident with sound offset. **e** Paired comparisons of mean evoked firing rate during control and laser trials. Paired $t$ tests yielded a significant increase in evoked firing rate during optogenetic suppression for clean ($n = 43$ configurations; $P = 4.50e-07$, $d = -0.91$) and masked trials ($n = 18$ configurations; $P = 0.006$, $d = -0.073$).

**Investigating cortical coding in mouse ACx using a cocktail party-like paradigm**. To better understand cortical coding of complex scenes in a mouse model amenable to circuit interrogation using genetic tools, we adopted a cocktail party-like experimental paradigm[25] while recording from neurons in ACx. Specifically, we recorded responses to spatially distributed sound mixtures to determine how competing sound sources influence the cortical coding of stimuli. The recording configuration consisted of four speakers arranged around the mouse at four locations on the azimuthal plane: directly in front (0°), two contralateral (45° and 90°) and 1 ipsilateral (−90°) to the right auditory cortex recording area. Target stimuli consisted of white noise modulated by human speech envelopes extracted from a speech corpus[29]. We utilized two target waveforms (target 1 and target 2) and a competing masker consisting of unmodulated white noise. Mice were exposed to either target-alone trials (Clean) or target-masker combinations (Masked) (Fig. 2a–c).

**Mouse ACx neurons show spatial configuration sensitivity between competing auditory stimuli**. We assessed cortical coding using neural discriminability, which refers to the ability to determine stimulus identity based on neural responses and, thus a neuron's ability to encode stimulus features, and a variety of other quantitative response measures. Neural discriminability between

the two targets (% correct) was computed both without the masker (Clean) (Fig. 2a); and with the masker (Masked), for all possible combinations of target and masker locations (Fig. 2b, c). We refer to the matrix of performance values from all speaker configurations as spatial grids, which allow for visualization of the spatial tuning sensitivity of a given unit in the presence of competing auditory stimuli (Fig. 2d). Values near 100 and 50%, respectively, represent perfect discriminability and chance discriminability, and positions of high performance (≥70%), which were also statistically significant ($P < 0.05$) with a relatively large effect size (Fig. 2e, $d \geq 1$), were deemed as hotspots. These hotspots represent locations of enhanced discriminability between the two targets, either in the absence (Clean) or presence (Masked) of a competing masking stimulus, using a spike-distance-based classifier to determine how well target identity can be predicted given the spike train from that site based on dissimilarities in spike timing and instantaneous rate[25] (see "Neural discriminability using SPIKE-distance").

Figure 2a illustrates spike trains from an example SU that shows high discriminability under both target-only conditions (Fig. 2a, d, e, black); and for a specific spatial configuration in the presence of a competing noise masker (Figs. 2b, d, e, red). In the masked condition, discriminability depends strongly on the spatial configuration of the target and masker, indicating that the

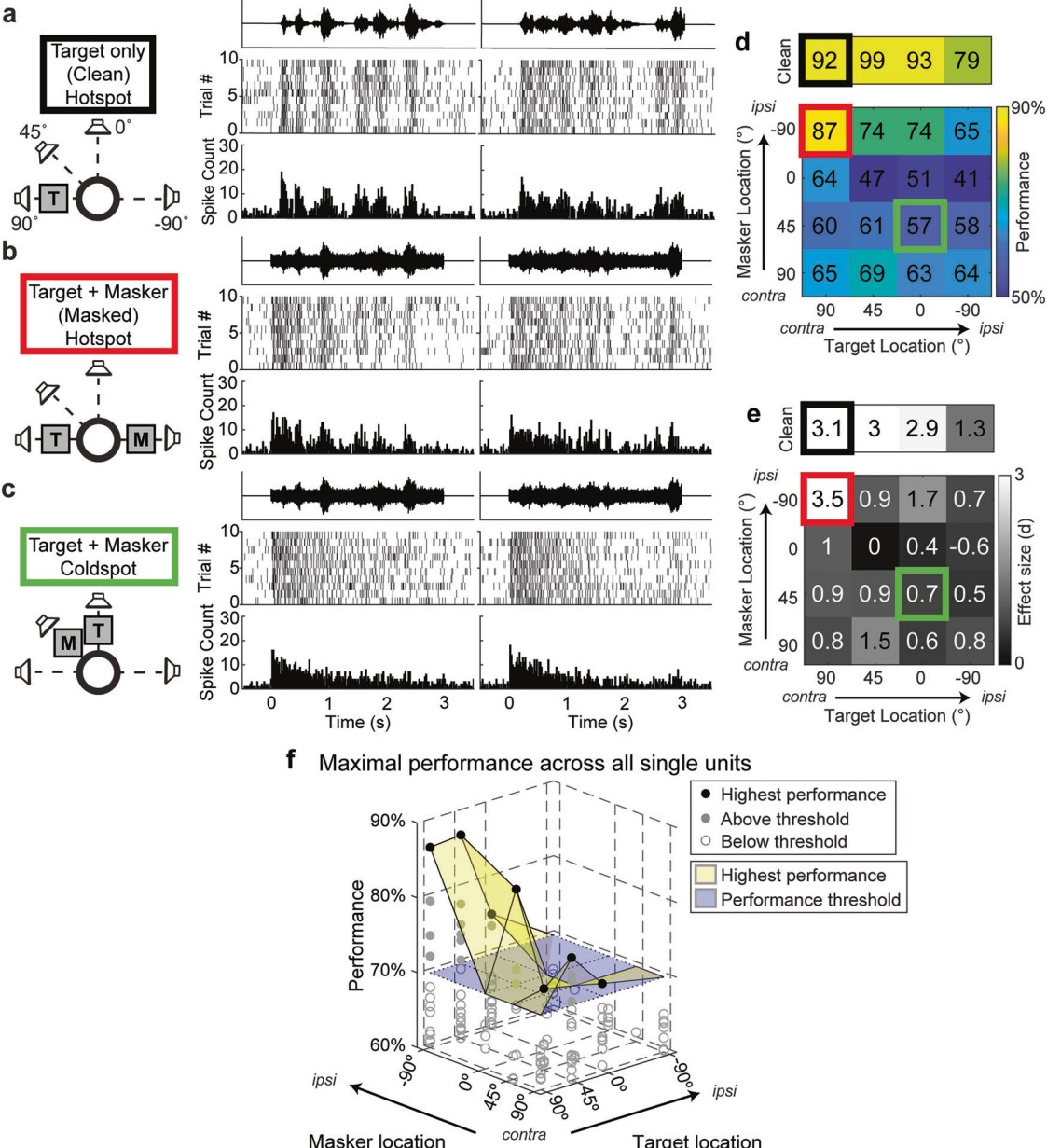

**Fig. 2 Cortical discrimination in a cocktail party paradigm in mouse ACx.** Auditory stimuli were presented from speakers at four locations. Target stimuli consisted of white noise modulated by human speech envelopes extracted from recordings of speech sentences (see "Auditory stimuli"). **a** Responses to both target stimuli (T) for clean trials originating at +90° azimuth. All plotted PSTHs have a bin length of 20 ms. During clean trials, responses exhibit spike timing and rapid firing rate modulation that follow the amplitude envelope of both target stimuli. **b** Responses during trials where targets (T) played at +90° and a competing masking stimulus (M) played at −90°. Masking stimuli consisted of unmodulated white noise with the same onset and offset times as target stimuli. In this configuration, spike timing and firing rate modulation follow both target stimuli, despite the presence of the competing masker. **c** Responses during trials where targets played at 0° and maskers played at +45°. For this configuration, spike timing and firing rate modulation do not follow either target stimulus, resulting in similar responses between target identities. **d** Neural discriminability performance for all possible target-masker location configurations, referred to as the spatial grid, for the example cell featured in (**a–c**). Outlined spots indicate configurations shown in (**a–c**), matched by the outline color. Performance is calculated using a template-matching approach based on differences in instantaneous firing rate and spike timing similarities between spike trains (see "Neural discriminability using SPIKE-distance"). The top grid shows discriminability for clean trials on top, while the bottom grid shows discriminability for masked trials. All blocks are color-coded according to the color axis shown to the right of the masked grid. Configurations with high performance (≥70%) and a large effect size (d ≥1), e.g., the configurations outlined in black and red, are referred to as hotspots. **e** Effect sizes for each spatial grid configuration in (**d**), with the same outlines corresponding to (**a–c**). Positive values represent an increase in performance relative to a null distribution where spike trains within each target are template-matched to each other, while negative values represent a decrease in performance relative to null. **f** The performance of all 19 single units exhibiting at least one hotspot during control trials. The translucent yellow surface represents the upper envelope of best performance across all single units for each masked spatial configuration, while the translucent blue surface represents the performance threshold of 70% for hotspots. Solid gray markers represent masked configurations with performances above the threshold, while unfilled gray markers represent data points with performances below the threshold. Black markers represent the maximal performance used to represent the upper envelope.

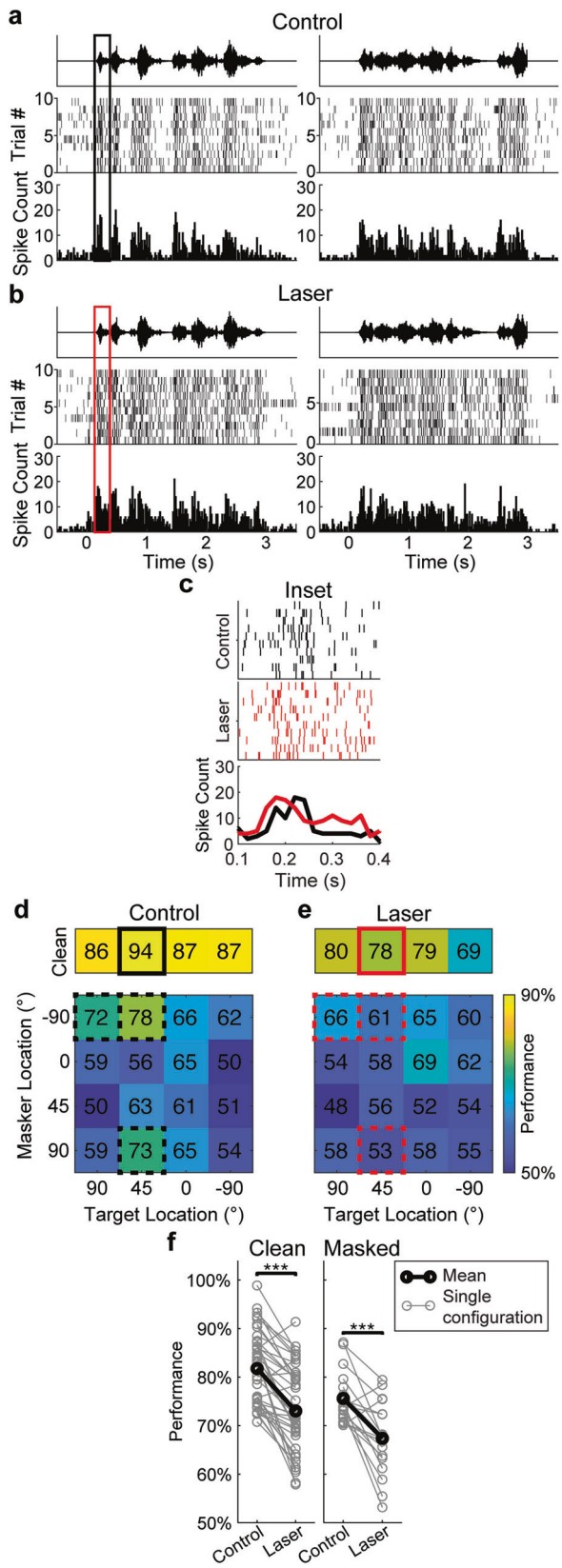

**Fig. 3 Effects of suppressing PV neurons on cortical discrimination.**
**a** Responses during clean stimulus trials originating at 45° from one example cell during control conditions. **b** Responses from the same cell and clean stimulus location in (**a**) during optogenetic conditions. **c** Inset showing zoomed-in portion of the response between 0.1 and 0.4 s after sound onset, as outlined in (**a**, **b**). Responses during optogenetic trials show earlier onset and reduced spike timing consistency, compared to the control. **d** Example spatial grid from the single unit in (**a–c**) during control conditions. **e** Example cell's spatial grid during optogenetic conditions, with the performances at Clean Target 45° outlined in black to correspond to responses shown in (**a–c**). Performance is color-coded according to the axis shown to the right of the Laser grid. The reduction in spike timing reproducibility during optogenetic suppression (seen in **c**) contributes to the decrease in performance (78%) compared to control trials at the same configuration (94%). In addition, performance decreased during optogenetic suppression for the rest of the clean configurations, while performance at the masked control hotspots, outlined by dashed boxes in both (**d**, **e**), decreased to below threshold: Target 45°, Masker 90° (73–53%); Target 45°, Masker −90° (78–61%); and Target 90°, Masker −90° (72–65%). **f** Paired comparisons of SPIKE-distance-based performance from control and PV-suppressed trials at the same spatial grid location. Paired *t* tests yielded a significant decrease in performance for both clean (*n* = 43 configurations; *P* = 3.44e-10, *d* = 1.24) and masked (*n* = 18 configurations; *P* = 2.50e-04, *d* = 1.09) trials during optogenetic suppression, indicating that PV suppression significantly reduced discrimination performance.

were curious to test how the performance of the best neurons in our population would be affected by optogenetic suppression of PV neurons. To do so, we focused on the neurons with high discrimination performance in our population (i.e., SUs with at least one hotspot in the clean or masked conditions). Nineteen SUs showed hotspots at one or more spatial configurations, and there were 43 hotspots in the clean condition and 18 hotspots in the masked condition giving a total of 61 hotspots.

At each spatial configuration, we observed a broad range of performance levels, consisting of neurons with significant hotspots (Fig. 2f, filled circles), as well as neurons with poor performance (open circles), reflecting that different neurons in the population had different spatial configuration sensitivities. The upper envelope of maximal performance was relatively high for all spatial configurations, except co-located target-masker and ipsilateral target positions. Thus, as a population, ACx neurons showed robust performance at all spatial configurations in the contralateral hemisphere, when the target and masker were spatially separated. We did not observe any statistically significant differences in performance between SUs across different layers, or SUs with different waveform types (RS vs. NS) (Supplementary Fig. 7).

**Suppression of PV neurons reduces discrimination performance at hotspots.** To investigate the role of PV interneurons in auditory discrimination performance, we compared discrimination performance at hotspots, with and without optogenetic suppression of PV neurons in ACx. Figure 3a–e shows an example SU with and without suppression. Compared to the control response (Fig. 3a), the optogenetic response (Fig. 3b) shows an increase in spiking between the peaks of both target stimuli. Specifically, the responses exhibited an earlier onset and decreased spike timing reproducibility across trials during suppression (Fig. 3c). Figure 3d shows the spatial grids for the same example SU during both conditions, with the example configuration in Fig. 3a, b (Clean Target 45°) outlined in black in the control grid (Fig. 3d) and in red in the optogenetic grid (Fig. 3e).

response of this neuron is spatial configuration-sensitive (Fig. 2b–e, red versus green).

Previous studies have demonstrated that neurons with the highest performance are most strongly correlated with behavior and strongly constrain population performance[30–34]. Thus, we

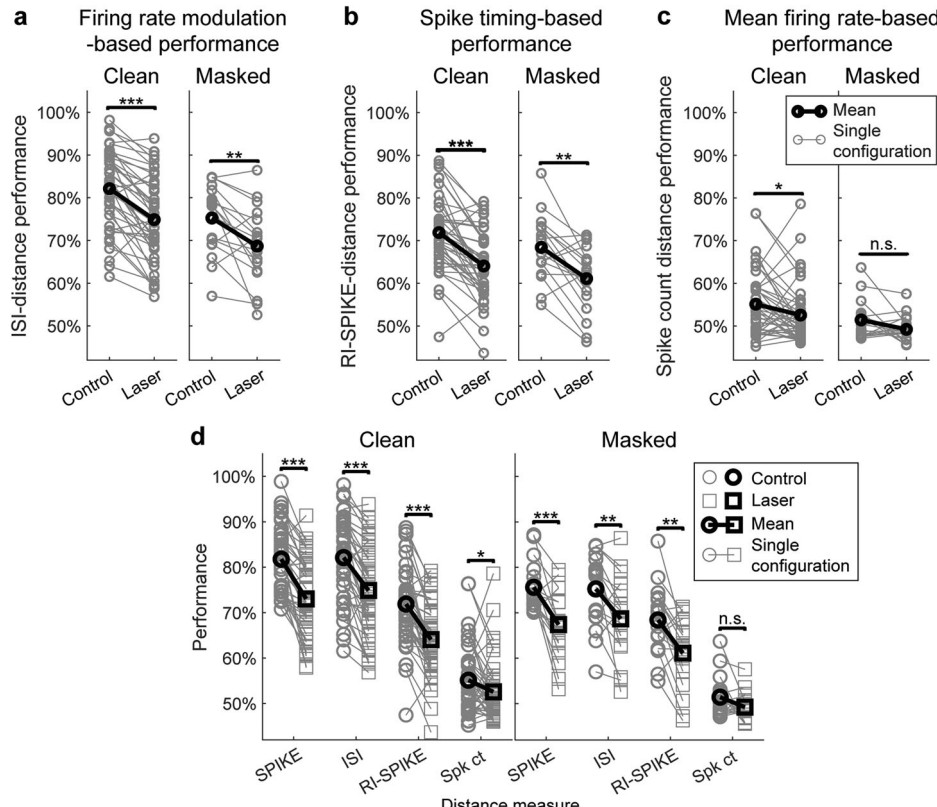

**Fig. 4 Effects of suppressing PV neurons on spike timing and rate-based coding measures. a** Performance based on ISI-distance, which measures differences between trains in instantaneous firing rate only (see "ISI-distance"). Paired $t$ tests showed a significant decrease in performance for both clean ($n = 43$ configurations; $P = 1.33\text{e-}08$, $d = 1.07$) and masked ($n = 18$ configurations; $P = 0.00371$, $d = 0.79$) trials. **b** Performance based on RI-SPIKE-distance, which measures differences between trains in spike timing only (see "RI-SPIKE-distance"). Paired $t$ tests showed a significant decrease in performance for both clean ($P = 2.63\text{e-}08$, $d = 1.04$) and masked ($P = 0.0010$, $d = 0.32$) trials. **c** Performance based on differences in total spike count between spike trains was near chance level, indicating that total spike count did not account for overall discrimination performance. Paired $t$ tests showed a significant decrease in performance for clean trials ($P = 0.0434$, $d = 0.32$) but not for masked trials ($P = 0.0656$, $d = 0.46$). **d** Summary figure showing contributions from spike-distance measures presented in Fig. 3f and panels (**a–c**) on the same scale and axis, with circle markers representing performance during control trials and square markers representing performance during optogenetic trials. Changes in spike timing and instantaneous firing rate-based measures (RI-SPIKE and ISI, respectively) provide relatively high discrimination performance and show a significant decrease upon optogenetic suppression of PV neurons.

This unit showed a decrease in performance across all clean configurations, and the hotspots in the control masked condition (Target 90°, Masker −90°; Target 45°, Masker −90°; Target 45°, Masker 90°) showed a reduction in performance to below threshold. Overall, we found that performance decreased significantly in both clean ($P = 3.44\text{e-}10$) and masked ($P = 2.5\text{e-}04$) conditions during suppression (Fig. 3f), a decrease that was not significant during laser stimulation in mice that did not express PV-Arch (Supplementary Fig. 8).

**Suppression of PV neurons degrades cortical temporal coding**. To determine the extent to which changes in the temporal dynamics of rapid firing rate modulation, spike timing, and average firing rate changes that occur during suppression might affect performance, we calculated different performance metrics across all hotspots. Specifically, we used inter-spike interval (ISI)-distance, rate-independent (RI)-SPIKE-distance, and spike count, as the basis for discriminability between spike trains. ISI distance calculates the distance between two spike trains based on the dissimilarities in instantaneous firing rate modulation, while RI-SPIKE-distance measures spike timing dissimilarity between two trains while accounting for changes in firing rate differences[21]. Spike count distance is the absolute difference in the number of spikes between trains, effectively measuring differences

in total firing rate. We found that performance based on both ISI-distance (Fig. 4a) and RI-SPIKE-distance (Fig. 4b) performances were relatively high. Both performances showed highly significant decreases with optogenetic suppression (ISI-distance-based performance: $P_{clean} = 1.33\text{e-}08$, $P_{masked} = 0.0037$; RI-SPIKE-distance-based performance: $P_{clean} = 2.63\text{e-}08$, $P_{masked} = 0.0010$). In contrast, performance based on spike count over the entire stimulus (Fig. 4c) was close to chance level both for control and laser conditions, indicating that spike count alone was not sufficient to account for overall performance. The significant decrease in ISI-distance-based performance indicates a disruption in rate-based coding, including the dynamics of instantaneous firing rate modulations. The significant decrease in RI-SPIKE-based distance indicates that spike timing-based coding is also degraded by optogenetic suppression of PV neurons (Fig. 4d).

**Effects on components of discrimination performance with suppression**. Generally, discrimination performance depends on both the dissimilarity of responses between targets, as well as the similarity of responses within a target. To assess the relationship between different components of responses with performance, we calculated three metrics sensitive to firing rate and/or timing: the average firing rate; the rate-normalized root-mean-square (RMS) difference in the responses to the targets,

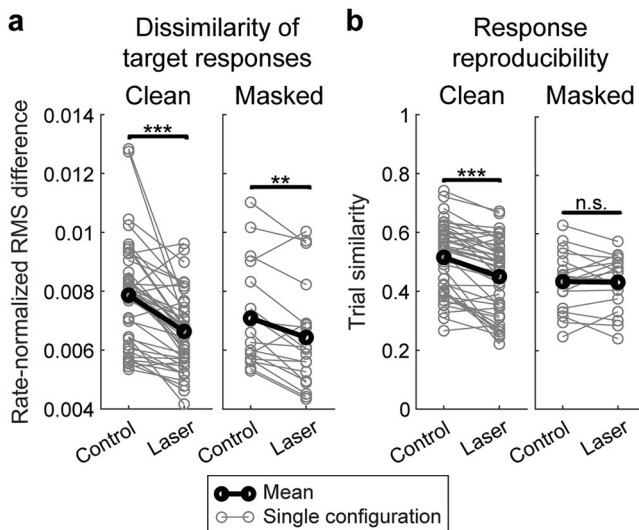

**a** Dissimilarity of target responses

**b** Response reproducibility

**Fig. 5 Effects of optogenetic suppression on spiking activity measures.**
**a** Changes in dissimilarity of target responses via rate-normalized RMS difference between target PSTHs during both conditions. Paired $t$ tests found significant decreases between conditions during both clean trials ($n = 43$ configurations; $P = 2.48e-07$, $d = 0.94$) and masked trials ($n = 18$ configurations; $P = 0.0024$, $d = 0.84$). **b** Changes in response reproducibility via trial similarity between responses to the same target during both conditions. Paired $t$ tests found a highly significant decrease between conditions during clean trials ($n = 43$ configurations; $P = 3.04e-08$, $d = 1.03$) but not masked trials ($n = 18$ configurations; $P = 0.823$, $d = 0.05$).

which captures the difference in the temporal pattern of responses to the targets; and the trial similarity[35], which captures the reproducibility of responses across trials within a target (see "Methods").

We first calculated the correlation between evoked firing rate and performance during the control condition by pooling clean and masked data. Firing rate did not show a significant correlation with performance ($r = -0.1286$, $P = 0.1582$), whereas both RMS difference and trial similarity measures were highly correlated with performance (rate-normalized RMS difference: $r = 0.4492$, $P = 2.11e-07$; trial similarity: $r = 0.5955$, $P = 4.69e-13$). These results suggest that both the pattern of firing rate modulations (quantified by RMS difference) as well as the reproducibility of responses (quantified by trial similarity) contribute to discrimination performance under control conditions. When comparing these measures between the control and laser conditions, we found that rate-normalized RMS difference significantly decreased with optogenetic suppression for both clean and masked trials (Fig. 5a), and trial similarity significantly decreased during clean trials (Fig. 5b).

**Optogenetic suppression decreases performance across a wide range of timescales.** The previous analyses used spike-distance measures which do not require a choice of a specific timescale for analysis. A further interesting question regarding discrimination performance is the optimal timescale for discrimination. Thus, we next quantified the timescale for optimal discrimination using the van Rossum spike-distance measure[36] (see "Methods").

We found that the optimal timescales for discrimination ($\tau$) for most neurons was around 40 milliseconds, with a significant proportion of neurons covering even finer timescales down to ~10 ms (Fig. 6a). Optimal $\tau$ was not significantly different

between control and laser conditions for clean trials ($P = 0.978$) but significant for masked trials ($P = 0.00315$), and performance decreased significantly in the laser condition across a wide range of timescales (Fig. 6b–d and Table 1). These results indicate that PV suppression did not significantly change the optimal timescale for discrimination but rather degraded discrimination across a wide range of timescales.

## Discussion

One of the most striking features of the cerebral cortex is the tremendous diversity of its cell types[37]. Understanding the computational role of such diversity in cortical coding is central to systems neuroscience. Addressing this central question requires understanding cell type-specific contributions to the cortical code at both the single neuron and population levels. A small number of previous studies have demonstrated a role of specific cell types in cortical population coding, specifically the generation of oscillations[38,39] and synchrony across cortical layers and areas[40,41]. However, cell type-specific contributions to the cortical code at the single-unit level, a fundamental aspect of cortical encoding, remain poorly understood. In this study, we addressed this fundamental gap by investigating the role of PV neurons in cortical coding of a complex scene, i.e., a cocktail party-like setting, in mouse ACx.

We assessed cortical coding using neural discrimination performance and other quantitative measures. There is a rich history of quantitative work on cortical discrimination[30,32]. These studies have suggested a critical role for neurons with the highest levels of performance in a population, which correlate strongly with behavioral performance and determine the overall performance at the population level. In this study we examined cortical discrimination of dynamic stimuli in a complex scene by the highest performing neurons in ACx, extending the previous body of work in several ways: First, we assessed the impact of optogenetic suppression of PV neurons on discrimination performance. Optogenetically suppressing PV neurons resulted in increased firing rate during spontaneous and auditory evoked activity, which is consistent with the effects of inhibitory blockade on cortical responses[42–44]. A recent study by Moore et al.[19] employed optogenetic suppression of PV neurons to powerfully reveal an important property of cortical networks: rapid rebalancing of excitation and inhibition upon PV suppression. Our study reveals that despite such rebalancing, cortical discrimination performance is degraded across cortical layers in ACx upon PV suppression. This finding suggests that PV neurons play a role in improving discrimination of dynamic stimuli in ACx, both sounds in quiet backgrounds, as well as in the presence of competing sounds from other spatial locations. Second, we quantified the contributions of instantaneous firing rate modulations, spike timing, and spike count towards cortical discrimination, using a family of spike-distance metrics. These metrics provide a powerful set of tools for dissecting different components of cortical coding. Although these metrics have been employed in previous theoretical studies, to our knowledge this is the first time they were applied to analyze cortical responses. This analysis revealed that high discrimination performance is mediated by the temporal pattern of firing rate modulations and spike timing reproducibility, and that optogenetic suppression of PV neurons degraded both components.

Previous studies have demonstrated that auditory cortical neurons can employ both rate and spike timing-based codes[3,4]; and provided insight into the roles of inhibitory neurons in shaping frequency tuning[11–13,45], frequency discrimination[14,46], adaptation[15], sparseness[47], and gap encoding[28]. An influential review on neural coding also defined a precise notion of a

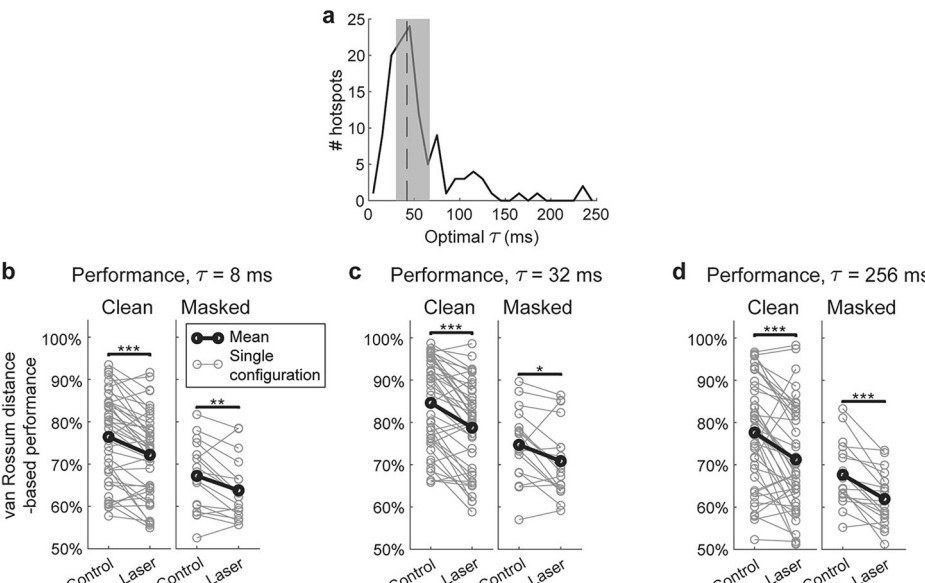

**Fig. 6 Decoding time analysis. a** Histogram of optimal $\tau$ for hotspots across both conditions (control and laser) and stimulus types (clean and masked). Dashed line indicates median value of 42ms, and shaded region represents the inter-quartile range (IQR) between 30ms and 67ms. Paired $t$-tests did not find a significant change in optimal $\tau$ within hotspots between conditions during clean trials ($n = 43$ configurations; $p = 0.9783$, $d = -0.00$) but found a significant decrease during masked trials ($n = 18$ configurations; $p = 0.00315$, $d = 0.81$). **b** van Rossum-based performance with $\tau$ set at 8ms. Performance was found to significantly decrease during both clean ($p = 4.65e-06$, $d = 0.80$) and masked ($p = 0.00392$, $d = 0.79$) trials. **c** van Rossum-based performance with $\tau$ set at 32ms. Performance was found to significantly decrease during both clean ($p = 6.99e-08$, $d = 1.00$) and masked ($p = 0.0152$, $d = 0.64$) trials. **d** van Rossum-based performance with $\tau$ set at 256ms. Performance was found to significantly decrease during both clean ($p = 5.10e-05$, $d = 0.69$) and masked ($p = 1.19e-04$, $d = 1.17$) trials.

temporal code as one that contains information in spike timing beyond rate modulations[48]. Temporal codes have been challenging to identify because the contributions of rate vs. spike timing are often difficult to decouple. Our results based on spike-distance measures, which quantify both rate-dependent and rate-independent components of coding, suggest that PV neurons specifically contribute to temporal coding in cortical discrimination. This computational portrait of PV neurons validates the importance of their established electrophysiological specializations–namely, fast, efficient, and temporally precise responses[5].

From a comparative standpoint, we found that the key features present at the cortical level within ACx of the mammalian mouse were consistent with previous findings in songbirds. Specifically, a previous study by Maddox et al. found hotspots at particular spatial configurations of target and masker on the spatial grids of cortical level neurons in songbirds[25]. Songbirds and mice have different frequency ranges of hearing and therefore the cues used for spatial processing, e.g., interaural time difference (ITD), and interaural level difference (ILD) are frequency-dependent, and the peripheral representations of these cues are likely to be different across species with different frequency ranges of hearing. This suggests the emergence of general cortical representations for solving the cocktail party problem despite different peripheral representations of acoustic cues across species.

A further interesting question regarding cortical discrimination is: what is the optimal timescale for maximal discrimination performance? One characteristic timescale in our stimuli arises from the slow modulation of speech envelopes on relatively long timescales ~100–500 ms, or equivalently, in the 2–10 Hz frequency range[49]. We found that the optimal timescale for most neurons in our dataset was much finer ~40 ms, with a significant number at even finer timescales down to ~10 ms. These timescales are well matched to the duration of short ultrasonic vocalizations in mice (Fig. 7), and finer grain structures within

**Table 1. Effect sizes and paired $t$-test results for all $\tau$ values used in van Rossum distance-based performance calculations**

| $\tau$ (ms) | $d_{clean}$ | $p_{clean}$ | $d_{masked}$ | $p_{masked}$ |
|---|---|---|---|---|
| 1 | 0.05 | 0.756 | 0.36 | 0.144 |
| 2 | 0.15 | 0.322 | 0.59 | 0.0220 |
| 4 | 0.39 | 0.0143 | 0.70 | 0.00882 |
| 8 | 0.80 | 4.65e-06 | 0.79 | 0.00392 |
| 16 | 0.97 | 1.10e-07 | 0.70 | 0.00852 |
| 32 | 1.00 | 6.99e-08 | 0.64 | 0.0152 |
| 64 | 0.88 | 9.40e-07 | 0.75 | 0.00573 |
| 128 | 0.78 | 7.23e-06 | 1.05 | 3.59e-04 |
| 256 | 0.69 | 5.10e-05 | 1.17 | 1.19e-04 |

these vocalizations, e.g., spectral features and frequency sweeps[50]. These timescales are similar to those found in a previous study of decoding sinusoidally amplitude-modulated (SAM) tones in mouse auditory cortex[35], and consistent with integration timescales in cat auditory cortex[51]. Phonemic structures in speech also occupy similar timescales, which are in the beta and low gamma range of frequencies[52,53]. Thus, the timescales for optimal discrimination in ACx, may be well-suited for analyzing such vocalizations and the finer spectro-temporal features within.

Our findings are also relevant in the context of cortical noise, which can have a profound impact on cortical codes[54]. We found that suppressing PV neurons did not change the optimal timescale for discrimination but rather degraded performance at a wide range of timescales (Fig. 6). In addition, we observed that suppression impacted specific components underlying discrimination: Most notably, PV suppression decreased the difference in the pattern of responses (quantified by RMS difference) between targets as well as the reproducibility of responses across trials (quantified by trial similarity). Taken together, these

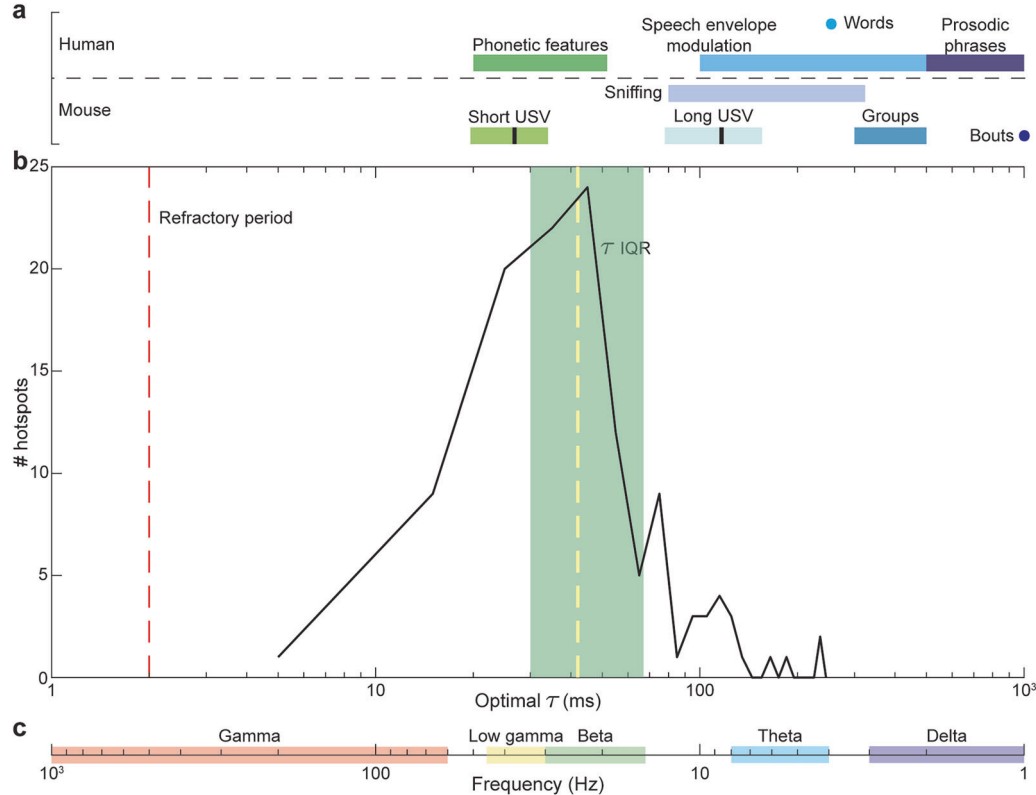

**Fig. 7 Comparison of optimal τ values and other timescales.** Semi-logarithmic plot showing various time-scales for spike timing in mouse ACx neurons compared to mouse vocalizations, human speech, and neural oscillations. **a** Time scales for human speech sounds, mouse vocalizations, and sniffing periods[79]. In the mouse time-scales, short and long USV bars represent the mean (black line) ± 2 SD. vocalization length. **b** Optimal τ plot reproduced from Fig. 6a (solid black line), with the red dashed line indicating the refractory period for single units, the dashed yellow line indicating the median value of 42ms, and the shaded green region representing the IQR. **c** Frequency bands for neural oscillations in Hz, decreasing from left to right.

observations are consistent with an overall enhancement in cortical noise level across multiple timescales upon PV suppression. A previous study on PV suppression in ACx observed the rapid rebalancing of excitation and inhibition, suggesting maintenance of the stability of global cortical representations[19]. However, our results suggest that despite this excitatory-inhibitory rebalancing, PV suppression also leads to an increase in cortical noise, fundamentally impacting the fidelity of cortical coding, including temporal coding.

Cortical inhibitory neurons can mediate feedforward, recurrent and di-synaptic feedback inhibition in cortical circuits (Supplementary Fig. 9). Previous modeling studies have demonstrated that feed-forward within-channel inhibition can improve discrimination performance[55]; whereas inhibition across different channels can lead to the formation of hotspots and the specific pattern of spatial configuration sensitivity[56]. Our results suggest that PV neurons mediate within-channel inhibition, corresponding to I neurons in the schematic model. This is consistent with our finding that although suppressing PV neurons reduced discrimination performance, it did not completely eliminate the presence of hotspots on the spatial grids, suggesting that PV neurons alone do not control the emergence of hotspots. Based on these observations, we hypothesize that a separate cell type (X neurons in Supplementary Fig. 9) mediates cross-channel inhibition, resulting in the generation of hotspots and the specific pattern of spatial configuration sensitivity on spatial grids. A candidate cell type that may correspond to X cells are somatostatin-positive (SOM) neurons, which have been implicated in di-synaptic feedback inhibition[57,58] and surround suppression[59,60]. These distinct roles may be functionally well-

suited for solving the cocktail party problem, with one class of neurons (PV) enhancing the temporal coding of dynamic stimuli at a target location, and another class of inhibitory neurons (X) suppressing competing stimuli from other spatial locations.

Several limitations in this study should be further addressed in the future. Although we used a cocktail party-like paradigm to probe auditory cortical responses to dynamic stimuli our experimental paradigm had some limitations. First, the target stimuli did not have any specific behavioral relevance, unlike the case of speech recognition at a cocktail party. Second, the masker stimuli did not contain any temporal modulations, unlike competing speakers at a cocktail party. Despite the anthropomorphic nature of our stimuli, we have demonstrated that auditory cortical neurons in mice are able to encode the distinct temporal features of both targets in the presence of a competing noise masker from different spatial locations. Future studies should address these limitations, e.g., by employing mouse communication sounds as targets and maskers. Although we were able to characterize the timescale for optimal discrimination in ACx, we did not characterize the integration window, or the encoding window[48,51,61]. Future studies that characterize both the timescale for optimal discrimination as well as the encoding window can address whether cortical neurons also employ temporal encoding, i.e., encode information in the temporal pattern of spikes within the encoding window[48]. Within this study, mice were awake but listening passively, whereas listening in a cocktail party-type setting is an active sensing process. It will be interesting to probe cortical coding in awake, behaving mice in experiments where animals attend to a specific spatial location. A recent theoretical model of attention in the auditory cortex, the AIM network

model[62], suggests distinct roles for different interneuron groups in attentional sharpening of both spatial and frequency tuning which enables flexible listening in cocktail party-like settings, e.g., monitoring the entire scene, selecting a speaker at a spatial location, and switching to a speaker at a different location. Future experiments probing distinct interneuron populations (e.g., PV, SOM, and VIP neurons) in behaving animals, in conjunction with testing and extending the AIM model, may further unravel cortical circuits for solving the cocktail party problem.

## Methods

**Subjects**. All procedures involving animals were approved by the Boston University Institutional Animal Care and Use Committee and the University of Illinois at Urbana-Champaign Institutional Animal Care and Use Committee (IACUC). A total of 14 C57BL6/J transgenic mice were used in this study. Original breeding pairs of parvalbumin-Cre (PV-Cre: B6;129P2-Pvalbtm1(cre)Arbr/J), and Ai40 mice (Arch: B6.Cg-Gt(ROSA)26Sor^tm40.1(CAG-aop3/EGFP)Hze/J) mice were obtained from Jackson Laboratory (Maine), and all breeding was done in house. Subjects consisted of both male and female PV-Arch ($n = 9$) offspring and PV-Cre ($n = 5$) only offspring (controls) 8–12 weeks old on the day of recording.

**Surgery**. Mice were surgically implanted with a headplate[63,64]. Briefly, under isoflurane anesthesia, stereotaxic surgery was performed to install a headplate, electrode, and optical fiber. The custom headplate was mounted anterior to the bregma allowing access to ACx caudally. The headplate was anchored to the skull with three stainless steel screws and dental cement. A fourth screw was connected to a metal pin and placed in the skull above the contralateral cerebellum to serve as the reference. A craniotomy was made above the right auditory cortex (AP −2.3 to −3.6, ML + 4.0 to +4.5, DV). Using a stereotaxic arm, a 32-contact linear probe (Neuronexus, Ann Arbor, MI; model: a 4 × 8–5 mm-100-400-177-CM32) with 100-µm spacing between electrode contacts and 400-µm spacing between shanks, was positioned into ACx, perpendicular to the cortical surface. Because of the curvature of the ACx surface, not all four shanks could be placed at precisely the same depth during each experiment. Probes were advanced until all electrode contacts were within the cortical tissue and shanks were positioned along the rostrocaudal axis of ACx (Fig. 1a–c). An optical fiber, 200 µm in diameter, was placed medially to the four shanks and positioned between the two innermost shanks terminating at the cortical surface (Fig. 1a). After implantation, mice were allowed to recover for 4–7 days before undergoing habituation to being head-fixed during recordings.

**Habituation**. Following surgery and complete recovery, mice were first handled for several days before being head-fixed to the recording apparatus. Mice were gradually exposed to longer restraint periods at the same time of day as subsequent recording sessions. Each animal received at least six habituation sessions prior to the first recording day. Under head-fixed conditions, mice were loosely covered with a piece of lab tissue taped down on either side (Kimwipes: Kimberly-Clark, Irving, TX) to encourage reduced movement. At the conclusion of habituation, mice underwent recording sessions in the presence of spatially distributed auditory stimuli.

**Recording sessions and data acquisition**. All recordings were made with a Tucker Davis Technologies (TDT; Alachua, FL) RZ2 recording system in an electrically-shielded sound attenuation chamber. Broadband neural signals at 24414.0625 Hz were recorded for each of the 32 channels. Local field potentials (LFPs) were band-pass filtered between 1 and 300 Hz, notch-filtered at 60 Hz, and digitized at 3051.8 Hz and used for current source density analysis.

Recording sessions consisted of both non-optogenetic and optogenetic trials in random order. The intertrial interval was 5 s, with 3 s of stimulus playback followed by 2 s of silence. Mice were exposed to target-alone (clean) trials and target-masker (masked) combinations. Ten trials were given per target identity for all possible combinations of target location, masker location (including clean trials), and optogenetic suppression of PV neurons. Thus, animals received a total of 800 trials per ~60 min recording session, with each session having a set laser power.

**Auditory stimuli**. All auditory stimuli were generated in Matlab and consisted of either target, masker, or combination of the two stimuli played from four TDT ES-1 electrostatic speakers. Target stimuli consisted of white noise modulated in time by human speech envelopes taken from the Harvard IEEE speech corpus[29] which has been used in previous psychological studies of the cocktail party effect[65]. Masker stimuli consisted of ten unique tokens of unmodulated white noise. Before speaker calibration, all stimuli were generated with the same RMS value, and sampling frequency was 195,312 Hz to capture the frequency range of hearing in mice. Stimuli were loaded onto a custom RPvdsEx circuit on an RZ6 Multi I/O processor, which was connected to two PM2R multiplexers that controlled the location of target and masker stimuli during trials.

During recordings, the stimuli were presented 18 cm from the mouse's head using four speakers driven by two TDT ED-1 speaker drivers. The four speakers were arranged around the mouse at four locations on the azimuthal plane: directly in front (0°), two contralateral (45° and 90°) and 1 ipsilateral (−90°) to the right auditory cortex recording area. Before recording sessions, stimulus intensity was calibrated using a conditioning amplifier and microphone (Brüel & Kjær, Nærum, Denmark; amplifier model: 2690, and microphone model: 4939-A-011). For 7 of the 9 Arch mice and the 5 PV-only control animals, all stimuli were at a measured 75 dB intensity at the mouse's head. For the remaining 2 Arch mice, stimulus intensity was set to 70 dB. Stimulus playback lasted 3 s with a 1 ms cosine ramp at onset and offset.

**Optogenetic stimulation**. Laser light for optogenetic stimulation of the auditory cortex was delivered through a multimode optically-shielded 200-µm fiber (Thorlabs, Newton, NJ; model: BFH48-200), coupled to a 532 nm DPSS laser (Shanghai Laser Ltd., Shanghai, China; model: BL532T3-200FC), with the fiber tip positioned right above the cortical surface. Laser power was calibrated to 2 mW, 5 mW, or 10 mW at the fiber tip using a light meter calibrated for 532 nm wavelength (PM100D, Thorlabs, Newton, NJ). The intensity was determined based on optogenetic cortical PV suppression studies using Archaerhodopsin from the literature[14,66]. During optogenetic trials, the laser was turned on 50 ms before stimulus onset and co-terminated with the end of the auditory stimuli (Fig. 1D). Square light pulses lasting 3.05 s were delivered via TTL trigger from the RZ2 recording system to the laser diode controller (ADR-1805). Optogenetic trials were randomized throughout the recording session such that animals received all stimulus/masker pairs from each location with and without laser. Recordings were done in successive blocks with constant optogenetic suppression strengths of 2 mW, 5 mW, or 10 mW, with each block lasting ~60 min and having their own set of control trials. These laser strengths are similar to those used in past studies[14,18] and did not result in epileptiform activity in the cortex.

**Histology**. At the end of the experiments, all mice were transcardially perfused, and tissue was processed to (1) confirm the specificity of ArchT expression to PV-cell populations within all PV-Arch animals and (2) confirm electrode placement in A1. Briefly, mice were perfused with 30 mL 0.01 M phosphate-buffered saline (Fisher Scientific, BP2944-100, Pittsburgh, PA), followed by 30 mL 4% paraformaldehyde (Sigma-Aldrich, 158127, St. Louis, MO). Brains were carefully removed and post-fixed 4–12 h in 4% paraformaldehyde before being transferred to a 30% sucrose solution for at least 24 h before sectioning. Brains were sectioned coronally at a thickness of 50 µm with a freezing microtome (CM 2000R; Leica) or cryostat (CM 3050 S; Leica). Tissue sections were collected throughout the auditory cortex. A subset of sections were stained with antibodies against PV (guinea pig anti-PV antibody, SWANT GP72 1:1000) followed by Alexa Fluor 568 goat anti-guinea pig secondary antibody (No: A-11075, Thermo Fisher Scientific, 1:500). Antibodies and dilution concentrations were previously reported[67–69]. Briefly, sections were rinsed with 0.01 M PBS followed by a solution of 100 mM glycine (No: G7126, Sigma-Aldrich) and 0.5% Triton-X in 0.01 M PBS. This was followed by a 2-h blocking buffer incubation with 5% normal goat serum and 0.5% Triton-X in 0.01 M PBS. Sections were then incubated for 24 h with primary antibody, rinsed with 100 mM glycine and 0.5% Triton-X in 0.01 M PBS, and incubated with secondary antibody for 2 hours. Slices were lastly incubated for 10 min with Hoechst 33342 (No: 62249, Thermo Fisher Scientific, 1:10,000 in 0.01 M PBS), rinsed with 100 mM glycine and 0.5% Triton-X in 0.01 M PBS before being rinsed in 100 mM glycine in 0.01 M PBS before mounting. Slices were mounted on slides (Fisherbrand Superfrost Plus, No: 12-5550-15, Fisher Scientific) using anti-fade mounting medium (ProLong Diamond, No: P36965, Thermo Fisher Scientific). For sections designated for imaging of electrode locations, tissue sections from the auditory cortex were mounted on gelatin-coated slides and allowed to dry overnight. Mounted sections were then rehydrated by being placed in deionized water for 5 min. Following rehydration, sections were incubated in 0.1% cresyl acetate for 5 min, followed by dehydration in an ascending series of alcohol rinses (50%, 70%, 90%, 95%, 100% (2×) for 3 min each and cleared with xylene (534056; Sigma-Aldrich, Natick, MA) for 15 min. Slides were then coverslipped using DPX (06522; Sigma-Aldrich, Natick, MA) mounting medium, allowed to dry, and imaged as described below.

**Imaging and quantification**. Images were taken on a VS120 wide-field Olympus microscope or an OlympusFV3000 scanning confocal microscope using a ×20 objective. All images were comprised of Z-stacks consisting of 5–6 slices taken at 10-µm intervals throughout the 50 µm slices. Stacks were taken from coronal sections as near as possible to the electrode location in the auditory cortex. Areas were chosen to include similarly dense Arch-GFP cell counts across animals. To confirm targeting specificity, each PV+ cell was categorized as co-expressing or not expressing Arch-GFP across a 300 × 300 µm grid. We also quantified the number of Arch+ cells from each stack that were not PV+ based on Hoechst labeling to estimate off-target expression. We analyzed 2–4 non-overlapping stacks from two slices per animal from the animals that made up the optogenetic Arch+ dataset ($n = 9$ PV-Arch). Cell counts were pooled across slices stained for the same marker for each animal and averaged to produce a single data point for quantification.

Nissl-stained sections taken from a subset of animals in the study ($n = 5$ mice) were taken with a Keyence BZ-X800E microscope using a 10X objective. Sections from each animal were stitched together, and electrode location was verified by overlaying images on drawings obtained from a digital stereotaxic atlas[70].

**Spike extraction and clustering**. Kilosort2 was used to detect multiunits within the recordings[27]. Before spike detection and sorting, the broadband signal was band-passed between 300 and 5000 Hz using a 3rd-order Butterworth filter. Kilosort results were then loaded onto Phy2 (https://github.com/cortex-lab/phy) to manually determine if spike clusters exhibited neural activity or noise[26]. Clusters with either artifact-like waveforms from laser or similar responses across all channels were deemed as noise, and spikes with artifact-like waveforms were removed from clusters that clearly exhibited neural activity, whenever possible. Clusters were merged if the cross-correlograms were similar to the component clusters' auto-correlograms and showed overlap in principal component feature space at the same channel. The spikes toolbox (https://github.com/cortex-lab/spikes) was then used to import the cluster information from Phy to Matlab and extract spike waveforms from the high-passed signal[27]. Clusters were assigned to recording channels based on which site yielded the largest average spike amplitude. To remove any remaining artifacts from laser onset and offset, all spikes with waveforms above an absolute threshold of 1500 μV or a positive value above 750 μV were discarded, and clusters that still showed a high amount of remaining artifact after removal were excluded from further analysis. To determine which of the remaining clusters were single units (SU), we utilized the sortingQuality toolbox (https://github.com/cortex-lab/sortingQuality) to calculate isolation distances and L-ratios[71]. SUs must (1) have less than 5% of inter-spike intervals below 2 ms (Fig. 1c), (2) an isolation distance above 15, and (3) an L-ratio below 0.25. For clusters where isolation distance and L-ratio were not defined, the first threshold was used. These thresholds are consistent with values used in past studies on single-unit activity[72–74], and clusters that did not meet any of these criteria were deemed multiunits (MUs). Finally, SUs were classified as narrow-spiking if the trough-peak interval of their mean waveform was below 0.5 ms, a threshold that is consistent with past findings on excitatory and inhibitory units in mouse auditory cortex[12].

**Current source density estimation and layer analysis**. Current source density (CSD) analysis estimates the second spatial derivative of LFP signals to determine the relative current across the cortical laminar depth. CSDs were calculated using LFPs, as described previously[64]. LFPs from control masked trials were used, as the rise time was more similar between target identities than clean stimuli. LFPs were low-passed and filtered to 150 Hz before being down-sampled by a factor of 8 to 381 Hz. For each channel, LFPs were averaged across all control masked trials prior to CSD estimation, and channels that did not show an evoked response were interpolated using neighboring sites on the same shank. After this, LFPs were spatially smoothed across the eight channels in each shank:

$$\phi(z) = \frac{\phi(z + \Delta z) + 2\phi(z) + \phi(z - \Delta z)}{4} \quad (1)$$

where z is the depth perpendicular to the cortical surface, Δz is the electrode spacing, and Φ is the potential. CSD was then estimated as:

$$CSD(z) = -\frac{\phi(z + \Delta z) - 2\phi(z) + \phi(z - \Delta z)}{\Delta z^2} \quad (2)$$

To determine the granular layer in each shank, CSD sink onset times were calculated as the time when the CSD goes below three times the standard deviation of pre-stimulus activity. If more than one channel was found to have the earliest sink onset, the channel whose neighbors had the earliest onsets was deemed the granular layer, or L4. The width of each layer was estimated based on previous anatomical studies[75]. L1 consisted of channels at least 500 μm above the input layer, L2/3 consisted of channels 200 μm to 400 μm above the channel with the earliest sink onset; L4 consisted of the input channel and the channel 100 μm above it; L5 consisted of channels 100 to 300 μm below the input layer, and L6 consisted of all channels at 400 μm below the input layer.

**Neural discriminability performance using SPIKE-distance**. Neural discrimination performance refers to the ability to determine stimulus identity based on neural responses, thus measuring a neuron's ability to encode stimulus features. Here, performance was calculated using a template-matching approach similar to our previous studies[25]. Briefly, spike trains were classified to one of the two target stimuli based on whose template, one from each stimulus, yielded a smaller spike distance. For each target-masker configuration, 100 iterations of template matching were done. In each iteration, one of the 10 spike trains for each target was chosen as a template, and all remaining trials were matched to each template to determine target identity. All possible pairs of templates were used across the 100 iterations to calculate an average value of neural discriminability. SPIKE-distance[21] calculates the dissimilarity between two spike trains based on differences in spike timing and instantaneous firing rate without additional parameters. For one spike train in a

pair, the instantaneous spike timing difference at time $t$ is:

$$S_1(t) = \frac{\Delta t_P^{(1)}(t) x_F^{(1)} + \Delta t_F^{(1)}(t) x_P^{(1)}}{x_{ISI}^{(1)}(t)}, t_P^{(1)} \leq t \leq t_F^{(1)} \quad (3)$$

where $\Delta t_P$ represents the distance between the preceding spike from train 1 ($t_P^{(1)}$) and the nearest spike from train 2, $\Delta t_F$ represents the distance between the following spike from train 1 ($t_F^{(1)}$) and the nearest spike from train 2, $x_F$ is the absolute difference between $t$ and $t_F^{(1)}$, and $x_P$ is the absolute difference between $t$ and $t_P^{(1)}$. To calculate $S_2(t)$, the spike timing difference from the view of the other train, all spike times and ISIs are replaced with the relevant values in train 2. The pairwise instantaneous difference between the two trains is calculated as:

$$S''(t) = \frac{S_1(t) + S_2(t)}{2\langle x_{ISI}^1(t), x_{ISI}^2(t)\rangle} \quad (4)$$

Finally, $S_1(t)$ and $S_2(t)$ are locally weighted by their instantaneous spike rates to account for differences in firing rate:

$$S(t) = \frac{S_1(t) x_{ISI}^2(t) + S_2(t) x_{ISI}^1(t)}{2\langle x_{ISI}^1(t), x_{ISI}^2(t)\rangle^2} \quad (5)$$

For a train of length $T$, the distance is the integral of the dissimilarity profile across the entire response interval, with a minimum value of 0 for identical spike trains:

$$D_S = \frac{1}{T} \int_0^T S(t) dt \quad (6)$$

cSPIKE, a toolbox used to calculate SPIKE-distance, was used to calculate all spike train distances between all possible spike train pairs for all spatial grid configurations[21].

To determine how firing rate modulation, spike timing, and average firing rate contribute to discriminability, we used different distance measures as inputs to the classifier. For all hotspots, performances using the inter-spike interval (ISI)-distance, rate-independent (RI)-SPIKE-distance, and spike count distance, the absolute difference in spike count between trains, were also calculated and compared to SPIKE-distance-based values.

**ISI distance**. To determine how optogenetic suppression affects rapid temporal modulations in firing rate, ISI-distances were calculated. The ISI distance calculates the dissimilarity between two spike trains based on differences in instantaneous rate synchrony. For a given time point:

$$I(t) = \frac{\left|x_{ISI}^{(1)}(t) - x_{ISI}^{(2)}(t)\right|}{\max\left(x_{ISI}^{(1)}(t), x_{ISI}^{(2)}(t)\right)} \quad (7)$$

This profile is then integrated along the spike train length to give a distance value, with values of 0 obtained for either identical spike trains or pairs with the same constant firing rate and a global phase shift difference.

**RI-SPIKE-distance**. To determine how optogenetic suppression affects spike timing, RI-SPIKE-distances between spike trains were calculated. The RI-SPIKE-distance is rate-independent, as it does not take differences in local firing rate between the two spike trains into account. From SPIKE-distance calculations, the final step of weighing $S_1(t)$ and $S_2(t)$ by their instantaneous spike rates is skipped, yielding:

$$S_{1,2}^{RI}(t) = \frac{S_1(t) + S_2(t)}{2\langle x_{ISI}^1(t), x_{ISI}^2(t)\rangle} \quad (8)$$

Like the other measures, the dissimilarity profile is integrated to give a distance value, with a value of 0 obtained for two identical spike trains.

**Rate-normalized RMS difference and trial similarity**. In addition to average firing rate, we also calculated two other measures to determine their impact on classification performance: the similarity of responses within the target, and the dissimilarity of responses across targets. To quantify the intertrial reliability of responses to target stimuli, we adopted the measure of trial similarity from previous studies[35]. Specifically, we randomly divided the ten trials in each configuration into two equal groups, binned spike times with a time resolution of 25 ms, and calculated Pearson's correlation coefficient between the two resulting PSTHs. This process was repeated 100 times to obtain a mean correlation coefficient, or trial similarity.

We also calculated the rate-normalized RMS difference between target responses to quantify the dissimilarity in the temporal pattern of responses between the two targets. We first binned each target response using the same time resolution as trial similarity (25 ms) and normalized each PSTH such that the sum of all bins over time was 1. The RMS difference between the two rate-normalized PSTHs was then calculated. This measure quantifies the dissimilarity in the temporal pattern of responses across the targets, accounting for differences in mean evoked firing rate between targets.

All three response measures (average firing rate, trial similarity, and rate-normalized RMS difference between targets) were correlated with SPIKE-distance-based performance using Pearson's correlation coefficients, with separate calculations done for control and laser trials.

**Decoding time analysis using van Rossum distances**. To estimate the decoding time of the spike trains at each hotspot, we used van Rossum distances[36]. Briefly, the van Rossum distance between two spike trains involves convolving each response with a decaying exponential kernel with time constant $\tau$. The distance between two smoothed spike trains $f_1(t)$ and $f_2(t)$ is calculated as:

$$D_{VR} = \sqrt{\frac{1}{\tau}\int_0^\infty \left(f_1(t) - f_2(t)\right)^2 dt} \qquad (9)$$

For each spatial grid configuration, a distance matrix containing the van Rossum distances between all possible spike train pairs was set as the input for the template-matching approach. Performance was calculated across a range of $\tau$ values, increasing in powers of 2 from 1 ms to 256 ms. Finally, to determine the optimal $\tau$ value at which performance was maximized for each configuration, we implemented a fine-grain parameter search where $\tau$ was varied in steps of 1 ms, with the optimization separately done for control and laser trials.

**Statistics and reproducibility**. All single units and spatial grid data were extracted from $n = 14$ subjects consisting of 9 PV-Arch-expressing mice and 5 non-Arch-expressing mice. Spatial grid hotspots of high neural discriminability were determined using three criteria: (1) mean performance must be above 70% during control trials; (2) mean control performance distribution must be significantly different from chance ($P < 0.05$), calculated using a null distribution obtained by classifying spike trains within each target, which should result in chance performance; and (3) the effect size given by Cohen's $d$ between the two distributions (control vs. null) must be greater than 1:

$$d = \frac{\bar{x}_1 - \bar{x}_0}{\sqrt{\frac{(n_1-1)s_1^2 - (n_0-1)s_0^2}{n_1 + n_0 - 2}}} \qquad (10)$$

where values with subscript 0 represent the mean, standard deviation, and number of template-matching iterations for the null performance distribution. In addition, configurations where at least three trials for one target showed zero spiking were excluded from analysis, to avoid inaccurate estimates of performance. This resulted in $n = 43$ clean configurations and $n = 18$ masked configurations, both of which were used to analyze the effects of suppression on discriminability and spiking activity. In the manuscript, we focus on SUs with hotspots in the control condition. We found a small number of emergent hotspots (11 from 10 single units across both clean and masked trials) where performance and effect size were both below threshold in the control condition but above threshold in the laser condition, with a median performance 71.4% and an interquartile range of 3.1%. To analyze the effects of suppression on performance metrics, we used built-in Matlab functions to run paired $t$ tests between control and optogenetic values to determine statistical significance ($P < 0.05$), with tests done separately for clean and masked trials.

We also analyzed low-performance hotspots—configurations with performances between chance and our threshold of 70%—in three separate groups: hotspots with effect sizes (1) between 0.2 and 0.5, (2) between 0.5 and 0.8, and (3) greater than 0.8. To determine whether low-performance hotspots showed similar changes in performance to our main set of hotspots, we ran paired $t$ tests and calculated the effect size of optogenetic suppression on discriminability. For the first group, we found 58 clean configurations and 244 masked configurations. Clean performance did not significantly decrease ($P = 0.134$, $d = -0.20$) while masked performance did ($P = 7.00e$-$11$, $d = 0.43$). For the second group, both clean ($n = 40$, $P = 0.030$, $d = 0.36$) and masked ($n = 119$, $P = 1.21e$-$15$, $d = 0.85$) performance decreased with suppression. For the last group, both clean ($n = 41$, $P = 0.00363$, $d = 0.49$) and masked ($n = 65$, $P = 2.11e$-$08$, $d = 0.79$) performance decreased with suppression.

For performance comparisons between layers and between narrow-spiking and regular-spiking units, we separately ran repeated-measures ANOVA and effect size calculations for clean and masked trials, with condition as the within-subjects factor and cell type (narrow-spiking or regular-spiking) or layer as the between-subjects factor. Effect sizes were calculated using the Measures of Effect Size toolbox (https://github.com/hhentschke/measures-of-effect-size-toolbox)[76], and post hoc Tukey–Kramer tests were carried out if the ANOVA returned significance. To estimate the effect of optogenetic suppression on spiking across layers, repeated-measures ANOVA was carried out for both spontaneous and onset firing rate. Spontaneous firing rate was defined as the average firing rate during the 50 ms between laser onset and sound onset, while onset firing rate was defined as the average rate during the first 0.5 s of sound stimulus playback. For both measures, ANOVA was done for regular-spiking single units, with layer as the between-subjects factor and condition as the within-subjects factor. Post hoc Tukey–Kramer tests were carried out if the ANOVA returned a significant factor or interaction. Repeated-measures ANOVA was not done for narrow-spiking single units due to the small sample size per layer[77].

**Reporting summary**. Further information on research design is available in the Nature Portfolio Reporting Summary linked to this article.

## Data availability
Source data for the figures are provided in Supplementary Data 1. The data used in this study are available upon reasonable request.

## Code availability
The custom code used for this study is available at Zenodo (https://doi.org/10.5281/zenodo.14290575)[78]. The version of Kilosort used in this study (2.0) is available at https://doi.org/10.5281/zenodo.4147288[27].

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

## Acknowledgements

This research was supported by the National Science Foundation (#1835270), the National Institutes of Health (#1R34NS111742-01 and #1T32DC013017-01A1), and the Boston University Micro and Nano Imaging Facility (NIH #1S10OD024993) We would like to thank Alberto Cruz-Martín, Michael Economo, and Conor Houghton for

comments and suggestions on the manuscript. We also thank Monty Escabí and Oded Ghitza for discussions on coding timescales in auditory cortex and speech.

## Author contributions

J.C.N. and H.J.G. performed all initial experiments. J.C.N. analyzed the data. N.M.J. provided technical support and code for analysis. R.A.M. and H.J.G. performed histological analysis. Z.Q. and H.J.G. performed additional experiments for control analysis. X.H. and K.S. obtained funding and supervised the study. J.C.N., H.J.G., X.H., and K.S. wrote the manuscript and contributed to the interpretation of the results.

## Competing interests

The authors declare no competing interests.
