## [Peer Review File · Communications Biology]

Reviewers' comments:

Reviewer #1 (Remarks to the Author):

The ms by Nocon et al. addresses the function of PV-expressing neurons (GABAergic inhibitory neurons targeting the soma of principal cells and other interneurons) in coding of information by performing multi-electrode recordings from the auditory cortex of mice in a cocktail-party like paradigm. To assess the function of the PV neurons, the author use transgenic animals PV-Cre crossed with a Cre-dependent inhibitory opsin (Archaeorhodopsin, Arch) and apply optogenetic suppression during the auditory paradigm. Single unit recordings are selected from the recordings with and without optogenetic suppression, the cells are sorted as principal cells and inhibitory neurons and their activity is examined using various "spike-distance metrics". The authors argue that their results demonstrate that PV neurons enhance cortical temporal coding and reduce cortical noise, and thereby improve "cortical representations of dynamic stimuli in complex scenes".

The main scientific question of the study is very exciting and highly timely, and the basic experimental design is state-of-the art. However, I feel that the results do not fully and convincingly support the authors' interpretation and conclusions.

First, and most importantly, there is no evidence that the optogenetic suppression is, indeed, working and producing the selective and significant reduction of PV activity. There were 101 single units identified, of which 9 were "narrow spiking", putative inhibitory interneurons. PV comprise 30-50% of the cortical inhibitory neuron population, however none of these cells were reported to reduce their activity during the photostimulation. Clearly, the sample is too small to reliably characterize if and how the activity of PV vs. non-PV interneuron activity is modulated.

Such data would be, on the one hand, an essential control to demonstrate the validity of the optogenetic approach. On the other, the authors would have a set of identified PV single unites and could analyze their activity in the control epochs in relation to that of the principal cells and also to other interneurons

Second, the paradigm is very anthropomorphic: it uses human speech samples but applied to mice. Also, the "masking signal" is not an interfering speech samples, but white noise. Thus, the paradigm seems to be more of a signal-to-noise based model with a non-physiological stimulus, than a cocktail-party scene.

Third, the physiological relevance of the "performance" measure and the spatial "hot spots" remain obscure and not easily accessible for the readers. While these parameters are, indeed, reduced during optogenetic stimulation, it is not clear if this is not a trivial consequence of an overall higher baseline activity, and as such a reduced signal-to-noise ratio.

Conversely, the relevance of the increase in spatial constellations with low performance values remains unacknowledged, without any explanation and discussion.

Minor points:

- What is the scientific relevance of the CSD analysis? This seems to be superfluous and can be fully omitted.
- line 118-9: "Optogenetic suppression occurred on 50% of trials" - Optogenetic suppression was randomly applied in approx. 50% of trials.
- Fig. 7 aims to conceptualize time scales. This I see misplaced here in an experimental study.
- Fig 8 is highly speculative with essentially no support from the results.

Reviewers #2-3 (Remarks to the Author):

Nocon and colleagues investigated the role of parvalbumin-expressing (PV) auditory cortex interneurons in complex scene analysis. Based on their combined electrophysiological and optogenetic experiments, the suppression of PV neurons disrupted cortical representations of dynamic stimuli in 'complex scenes' where white noise modulated by human speech was masked by unmodulated white noise. The Authors concluded that PV neurons can enhance temporal coding and reduce cortical noise to improve the cortical representation of complex scenes.

On one hand, these findings are novel and interesting, supported by a set of elegant analysis approaches. On the other hand, I have major methodological concerns regarding the experiments that underlie these results. I think these need to be convincingly addressed to prove that the analyzed data support the Authors' conclusions. If all controls support the claims and methodological shortcoming can be overcome, then this could be an interesting paper for the readers of Communications Biology.

Major points

1. The optogenetic inhibition experiments are lacking control animals not expressing Arch in PV neurons ('YFP' controls), which are necessary to rule out non-specific effects of illumination. No images have been presented to demonstrate opsin expression and its specificity, and to demonstrate the placement of the optic fibers and the electrodes (histology). Without these, it is uncertain whether the experiments have been interpreted correctly.
2. Sorting spikes into putative single neurons was not convincingly demonstrated, especially in light of the fact that the Authors used linear silicon probes with a relatively large inter-electrode distance, which are not optimized for recording single-cell activity. Cluster quality is only discussed in the methods section to the extent that 'artifact-like spike waveforms were deemed as noise'. The Authors should present evidence on the validity of single-units, including qualitative and quantitative measures of cluster quality. Kilosort output typically needs post-hoc curation to avoid split clusters and other forms of 'overclustering'. Were auto- and crosscorrelations considered and if yes, how were they taken into account? What was the distribution of isolation distance and/or L-ratio values? This seems important as cluster contaminations can possibly influence the quantitative coding variables.
3. It would make the interpretation significantly more convincing if the Authors could show the suppression of firing in identified PV interneurons. However, I don't consider this absolutely necessary.

Minor points

1. Readability could be improved by better explaining the rationale of each experiment and analysis, better explaining the specific terms like 'spatial grid', 'hotspots', 'masker', different coding variables, etc. More labels in the figure would also help.
2. Out of the 9 putative interneurons, how many showed 'hotspots'?
3. Was there a difference in the effect of illumination with the three levels of laser intensity used?

I believe that openness and transparency can increase the fairness of peer review. Therefore, I decided to sign my reviews.

Balazs Hangya

Reviewer #4 (Remarks to the Author):

Previously, Moore et al. (Neuron 2018) demonstrated that optogenetic suppression of PV+ neurons in the auditory cortex increases feedforward excitation as well as downstream PV+ neuronal firing. Building on this previous study, the authors have addressed how optogenetic suppression of PV+ neurons affects temporal coding. The authors have reported that optogenetic suppression of auditory cortical PV+ neurons degrades the discriminability of dynamic sounds, and this is attributed to changes in the temporal structure of spike trains.

Although the results are intriguing, there are several technical, but major issues highlighted below. Unless addressing these critical points, the results cannot be interpreted properly.

Before analyzing auditory-evoked responses, the following points need to be addressed.

Firstly, the authors must histologically assess whether Arch was expressed in PV+ cells, and what fraction of PV+ cells expressed Arch.

Secondly, the authors have not demonstrated any evidence of optogenetic "suppression" of PV+ cells. Although the authors have argued that their results are consistent with the previous results, Moore et al. (2018) clearly showed that Arch activation did hyperpolarize PV+ cells (see their Fig. S1). Thus, the authors' argument is not correct.

Thirdly, a crucial control experiment is missing. The authors must demonstrate that optical illumination alone cannot modulate any neuronal responses without expressing Arch by performing the same experiments in either PV-Cre or Ai40 mice. A long illumination could affect the temperature of the brain tissue and/or optical illumination could stimulate the retina, which can lead to non-specific, non-auditory activity. The authors must exclude these possibilities experimentally.

Finally, the authors used a silicon probe with 100 um spacing. Although "single units" were defined based on their ISI violation, this criterion is not sufficient. Like many other published papers, an additional measurement, such as isolation distance, is required to quantify how well-separated each cluster is from the rest of clusters.

We thank the reviewers for their detailed comments and suggestions on our manuscript.

We have now collected significant additional data and made major revisions to the manuscript to address all of the concerns noted by the reviewers.

Most importantly, we now provide new analysis with evidence of optogenetic suppression of PV+ cells. We have also performed new control experiments in non-Arch-expressing mice to rule out any non-specific effects.

Additionally,

- We have histologically assessed Arch expression in the PV+ cells within the mice used in our main analysis, demonstrating that these mice indeed express Arch in PV+ cells.
- We have added additional criteria for determining single units (isolation distance and L-ratio). We found that the main results from our analysis remain similar with the refined population of single units.
- We have added a Statistics and reproducibility section.

We address all of the reviewers' comments, point by point, below.

Reviewers' comments:

Reviewer #1 (Remarks to the Author):

The ms by Nocon et al. addresses the function of PV-expressing neurons (GABAergic inhibitory neurons targeting the soma of principal cells and other interneurons) in coding of information by performing multi-electrode recordings from the auditory cortex of mice in a cocktail-party like paradigm. To assess the function of the PV neurons, the author use transgenic animals PV-Cre crossed with a Cre-dependent inhibitory opsin (Archaeorhodopsin, Arch) and apply optogenetic suppression during the auditory paradigm. Single unit recordings are selected from the recordings with and without optogenetic suppression, the cells are sorted as principal cells and inhibitory neurons and their activity is examined using various "spike-distance metrics". The authors argue that their results demonstrate that PV neurons enhance cortical temporal coding and reduce cortical noise, and thereby improve "cortical representations of dynamic stimuli in complex scenes".

The main scientific question of the study is very exciting and highly timely, and the basic experimental design is state-of-the art. However, I feel that the results do not fully and convincingly support the authors' interpretation and conclusions.

Major points

(1) First, and most importantly, there is no evidence that the optogenetic suppression is, indeed, working and producing the selective and significant reduction of PV activity. There were 101 single units identified, of which 9 were "narrow spiking", putative inhibitory interneurons. PV comprise 30-50% of the cortical inhibitory neuron population, however none of these cells were reported to reduce their activity during the photostimulation. Clearly, the sample is too small to reliably characterize if and how the activity of PV vs. non-PV interneuron activity is modulated. Such data would be, on the one hand, an essential control to demonstrate the validity of the optogenetic approach. On the other, the authors would have a set of identified PV single unites and could analyze their activity in the control epochs in relation to that of the principal cells and also to other interneurons	For the 9 identified narrow-spiking (NS) single-units in our main dataset, we have now performed new analysis of their spiking activity with high temporal resolution after laser onset. Out of these 9 units, 8 showed a significant decrease in spiking during optogenetic suppression compared to the control data in the first 2 ms after laser onset. These same units also show increased spiking activity ~10ms after laser onset, likely arising from decreased inhibitory activity within the cortical network. This new analysis provides evidence of direct and fast optogenetic suppression, followed by indirect network effects, consistent with previous studies, e.g., Moore et al., 2018. To further confirm that suppression occurs within PV-Arch neurons, we ran a similar analysis on the NS units within 5 control non-Arch-expressing mice. We identified 13 NS single units from this control group, and the average response during light presentation did not show a significant change in spiking during the first 2 ms of laser onset. Individually, 12 of these 13 NS units did not show a significant change in spiking with laser. The results of this new analysis on narrow-spiking units in both PV-Arch and light-only control subjects are shown in Supplemental Figure 5.
--	--

(2) Second, the paradigm is very anthropomorphic: it uses human speech samples but applied to mice. Also, the “masking signal” is not an interfering speech samples, but white noise. Thus, the paradigm seems to be more of a signal-to-noise based model with a non-physiological stimulus, than a cocktail-party scene.	Our targets stimuli used a broadband carrier with slow temporal amplitude modulations, which is common in both speech and animal vocalizations (e.g., Singh and Theunissen, 2003; Ding et al 2017). Our paradigm of using competing sounds from different spatial locations is another prominent aspect of a cocktail party scene. To our knowledge, this is the first study to apply such a paradigm in mice. We demonstrated that auditory cortical neurons were able to encode the distinct temporal features of both target stimuli in the presence of a competing noise masker from different spatial locations. We believe these results are relevant to cocktail party scene encoding. We agree that it would be interesting to investigate paradigms using more complex targets and maskers, e.g., mouse vocalizations. We now note limitations of our stimuli in the context of a cocktail party scene and suggest this as a future direction to explore in Discussion.
(3) Third, the physiological relevance of the “performance” measure and the spatial “hot spots” remain obscure and not easily accessible for the readers. While these parameters are, indeed, reduced during optogenetic stimulation, it is not clear if this is not a trivial consequence of an overall higher baseline activity, and as such a reduced signal-to-noise ratio. Conversely, the relevance of the increase in spatial constellations with low performance values remains unacknowledged, without any explanation and discussion.	We have now clarified the physiological relevance of the performance measure further to make it accessible to readers (Lines 173-5 and 550-1). To confirm that the reduction in performance during optogenetic suppression of PV+ neurons was simply not due to increased baseline activity, we ran simulations on a computational model of auditory cortex where network noise was increased while PV+ activity remain unchanged. Our simulations showed that simply increasing baseline activity within the network did not produce the same magnitude of performance reduction with PV+ suppression. We now comment on the effects of optogenetic suppression on performance at “spatial constellations with low performance values” (Lines 653-63).
Minor points	
(4) What is the scientific relevance of the CSD analysis? This seems to be superfluous and can be fully omitted.	The scientific relevance of current source density analysis is to estimate the layer location of each single unit in our experimental data. Our analysis on whether the layer of single units affected discrimination

	performance did not yield layer as a significant factor. These results along with the current source density methods and analysis are now included in the Supplemental Material for completeness.
(5)	line 118-9: "Optogenetic suppression occurred on 50% of trials" - Optogenetic suppression was randomly applied in approx. 50% of trials. This sentence has been edited per comments at line 127 and 854 in the revised manuscript.
(6)	Fig. 7 aims to conceptualize time scales. This I see misplaced here in an experimental study. Figures 7 is included in the discussion section to visualize how the results from this study relate to time scales of neural discriminability, to help put our results in a broader context of the literature for the readers.
(7)	Fig 8 is highly speculative with essentially no support from the results. As with Figure 7, Figure 8 is meant to put the results in a broader context and indicate important future experiments and modeling work that would be necessary to test the overall hypothesis. Specifically, our working hypothesis is that multiple types of interneurons within cortex serve distinct functions in solving the cocktail party problem. The results in this manuscript support a role of PV neurons in improving temporal processing of dynamic stimulus features, consistent with models of within-channel feedforward inhibition (e.g., Wehr and Zador 2003; Narayan, Ergun and Sen, 2005). We hypothesize that another interneuron type, e.g., SOM neurons, could mediate "surround inhibition" across different spatial channels, which can be experimentally tested in future studies.

Reviewers #2-3 (Remarks to the Author):

Nocon and colleagues investigated the role of parvalbumin-expressing (PV) auditory cortex interneurons in complex scene analysis. Based on their combined electrophysiological and optogenetic experiments, the suppression of PV neurons disrupted cortical representations of dynamic stimuli in 'complex scenes' where white noise modulated by human speech was masked by unmodulated white noise. The Authors concluded that PV neurons can enhance temporal coding and reduce cortical noise to improve the cortical representation of complex scenes.

On one hand, these findings are novel and interesting, supported by a set of elegant analysis approaches. On the other hand, I have major methodological concerns regarding the experiments that underlie these results. I think these need to be convincingly addressed to prove that the analyzed data support the Authors' conclusions. If all controls support the claims and methodological shortcoming can be overcome, then this could be an interesting paper for the readers of Communications Biology.

Major points

(8) The optogenetic inhibition experiments are lacking control animals not expressing Arch in PV neurons ('YFP' controls), which are necessary to rule out non-specific effects of illumination. No images have been presented to demonstrate opsin expression and its specificity, and to demonstrate the placement of the optic fibers and the electrodes (histology). Without these, it is uncertain whether the experiments have been interpreted correctly.

To confirm that the main results in our analysis did not arise from non-specific effects of illumination, we included an additional control group with recordings from 5 PV-only littermate mice that did not express Arch in PV interneurons. These recordings used the same laser intensity (10mW) and stimulus intensity (75 dB) as those using Arch-expressing mice.

Across these 5 mice, we obtained 21 single-units that showed at least one spatial grid hotspot, resulting in a total of 46 clean and 40 masked hotspots. Using the same statistical tests, we found that there was no significant change in discriminability during both clean ($p = 0.0514$, $d = 0.30$) and masked ($p = 0.0689$, $d = 0.30$) trials. The effect size is also small, given the low values of Cohen's d , compared to the results from Arch-expressing mice, where d is close to 1 in both clean and masked trials.

When we looked at evoked firing rate, we found a significant decrease in spiking during clean trials ($p = 0.0482$, $d = 0.30$) but not during masked trials ($p = 0.2994$, $d = -0.28$). Once again, the significance and effect size of the clean results are small compared to the results from optogenetic mice. With these findings, our main results strongly support the conclusion that the effects arise from optogenetic suppression, and not non-specific effects of illumination. The results from these

	control experiments are displayed in Supplemental Figure 7. Also, we have since processed all the tissue from the PV-Arch mice recorded in the study. We quantified Arch expression specificity relative to immunohistological PV+ labeled neurons and Hoechst labeling. Our results indicate very high specificity of targeting only to PV cells as shown in the new Supplemental Figure 3.
(9) Sorting spikes into putative single neurons was not convincing demonstrated, especially in light of the fact that the Authors used linear silicon probes with a relatively large inter-electrode distance, which are not optimized for recording single-cell activity. Cluster quality is only discussed in the methods section to the extent that 'artifact-like spike waveforms were deemed as noise'. The Authors should present evidence on the validity of single-units, including qualitative and quantitative measures of cluster quality. Kilosort output typically needs post-hoc curation to avoid split clusters and other forms of 'overclustering'. Were auto- and crosscorrelations considered and if yes, how were they taken into account? What was the distribution of isolation distance and/or L-ratio values? This seems important as cluster contaminations can possibly influence the quantitative coding variables.	In our analysis, we manually curated the Kilosort results using Phy2, a graphical user interface for spike sorting information. We have made this manual step clearer within our Methods section. As for the question on auto- and cross-correlations, we utilized them to merge spike clusters. If cross-correlations between spike clusters were similar to the component clusters' auto-correlations, they were merged. These steps are now outlined in the Methods section as well. We have also added L-ratio and isolation distance as additional criteria for sorting single units. From the 'Spike extraction and clustering' section in Methods, lines 539-45: To determine which of the remaining clusters were single-units (SU), we utilized the sortingQuality toolbox (https://github.com/cortex-lab/sortingQuality) to calculate isolation distances and L-ratios⁶⁵. SUs must 1) have less than 5% of inter-spike intervals below 2ms (Figure 1C), 2) an isolation distance above 15, and 3) an L-ratio below 0.25. These thresholds are consistent with values used in past studies on single-unit activity^{66, 67, 68} including a recent paper that used the same 32-channel silicon probe, and clusters that did not meet any of these criteria were deemed multi-units (MUs). These additional criteria resulted in 82 out of our 124 identified units being deemed as SU, compared to the former value of 101. The major results of our previous analysis remain the same, and the manuscript is now edited to reflect the new sample size and p-values.

(10) It would make the interpretation significantly more convincing if the Authors could show the suppression of firing in identified PV interneurons. However, I don't consider this absolutely necessary.	From the 9 identified NS single-units in our main dataset, we analyzed their spiking during the first 2 ms after laser onset and compared mean firing rate between conditions. Out of these 9 units, 8 showed a significant decrease in spiking during optogenetic suppression compared to the control data in non-Arch-expressing mice where no significant suppression is observed. These results are now included in Supplemental Figure 5.
Minor points	
(11) Readability could be improved by better explaining the rationale of each experiment and analysis, better explaining the specific terms like 'spatial grid', 'hotspots', 'masker', different coding variables, etc. More labels in the figure would also help.	To address this comment, we have added more clarification on these terms when introduced in the main text. Specifically: We now provide a clearer explanation of the spatial grid, along with discriminability in lines 168-71: We refer to the matrix of performance values from all speaker configurations as spatial grids, which visualize the spatial tuning of a given unit in the presence of competing auditory stimuli. Values near 100% and 50% respectively represent perfect discriminability and chance discriminability... We also now better explain the meaning behind clean and masked trials in lines 173-5: These hotspots represent target locations of enhanced discriminability between the two targets, either in the absence (Clean) or presence (Masked) of a competing masking stimulus... We have also updated Figures 2, 3, 4, and 5 to improve their readability and better show what relevant features of neural responses were measured using our various metrics. These revised figures are attached below at the end of this reply document.
(12) Out of the 9 putative interneurons, how many showed 'hotspots'?	Out of the 9 putative interneurons, 5 showed at least one hotspot in either the clean or masked condition.
(13) Was there a difference in the effect of illumination with the three levels of laser intensity used?	We analyzed the effect of illumination intensity on the change in performance, spontaneous firing rate, and mean firing rate during stimulus playback between conditions. We restricted

this analysis to single-units, which were sorted using the refined criteria described in our reply to comment #9. For all analyses, we utilized a 1-way ANOVA on the difference between control and optogenetic conditions with intensity as the between-groups factor. Post-hoc multiple comparisons were carried out if the ANOVA yielded significance.

For both evoked and spontaneous firing rate, we found a significant effect of laser intensity on the magnitude of difference between conditions. Specifically, we found that the increases in evoked and spontaneous firing rate was significantly higher during 10mW recording blocks than in 2mW recording blocks ($p < 0.01$ for both measures).

Finally, we found that all laser powers resulted in decreased performance relative to the control condition. Between laser intensities, we found that the decrease in performance during the 10mW block was significantly larger than that of the 2mW ($p < 0.01$) and 5mW blocks ($p < 0.01$). Thus, our main analysis only includes the 10mW experiments, as the largest effects to occurred at that intensity.

Reviewer #4 (Remarks to the Author):

Previously, Moore et al. (Neuron 2018) demonstrated that optogenetic suppression of PV+ neurons in the auditory cortex increases feedforward excitation as well as downstream PV+ neuronal firing. Building on this previous study, the authors have addressed how optogenetic suppression of PV+ neurons affects temporal coding. The authors have reported that optogenetic suppression of auditory cortical PV+ neurons degrades the discriminability of dynamic sounds, and this is attributed to changes in the temporal structure of spike trains.

Although the results are intriguing, there are several technical, but major issues highlighted below. Unless addressing these critical points, the results cannot be interpreted properly. Before analyzing auditory-evoked responses, the following points need to be addressed.

Major points

(14) Firstly, the authors must histologically assess whether Arch was expressed in PV+ cells, and what fraction of PV+ cells expressed Arch.	As requested by two reviewers, we have processed all the tissue from the PV-Arch mice recorded in the study. We quantified Arch expression specificity relative to immunohistological PV+ labeled neurons and Hoechst labeling. Our results indicate very high specificity of Arch targeting only to PV cells as shown in the new Supplemental Figure 3.
(15) Secondly, the authors have not demonstrated any evidence of optogenetic “suppression” of PV+ cells. Although the authors have argued that their results are consistent with the previous results, Moore et al. (2018) clearly showed that Arch activation did hyperpolarize PV+ cells (see their Fig. S1). Thus, the authors’ argument is not correct.	We now provide evidence of optogenetic suppression of PV+ cells (Supplementary Figure 5). Out of the 9 NS units, 8 showed a significant decrease in spiking during optogenetic suppression compared to the control data in the first 2 ms after laser onset. These same units also show increased spiking activity ~10ms after laser onset, likely arising from decreased inhibitory activity within the cortical network. This new analysis provides evidence consistent with direct and fast optogenetic suppression, followed by indirect network effects, consistent with Moore et al., 2018. We note that Figure S1 in the Moore and Wehr paper shows recordings from auditory cortex slices, while our data are recorded in vivo. Thus, the magnitudes of direct optogenetic suppression and indirect network effects may be different, because of differences in network connectivity in slices vs. in vivo. In addition, effects of optogenetics resulted in a substantial increase in activity in all cells recorded relative to the non-laser condition that was not-present in our PV-Arch negative mice as described below.
(16) Thirdly, a crucial control experiment is missing. The authors must demonstrate that optical illumination alone cannot modulate any neuronal	We agree and to confirm that the main results in our analysis did not arise from non-specific effects of illumination, we ran additional recordings in 5 mice that did not express Arch

responses without expressing Arch by performing the same experiments in either PV-Cre or Ai40 mice. A long illumination could affect the temperature of the brain tissue and/or optical illumination could stimulate the retina, which can lead to non-specific, non-auditory activity. The authors must exclude these possibilities experimentally.	in PV interneurons. These recordings used the same laser intensity (10mW) and stimulus intensity (75 dB) as those using Arch-expressing mice. Across these 5 mice, we obtained 21 single-units that showed at least one spatial grid hotspot, resulting in a total of 46 clean and 40 masked hotspots. Using the same statistical tests, we found that there was no significant change in discriminability during both clean ($p = 0.0514$, $d = 0.30$) and masked ($p = 0.0689$, $d = 0.30$) trials. The effect size is also small, given the low values of Cohen's d compared to the results from Arch-expressing mice, where d is close to 1 in both clean and masked trials. When we looked at evoked firing rate, we found a significant decrease in spiking during clean trials ($p = 0.0482$, $d = 0.30$) but not during masked trials ($p = 0.2994$, $d = -0.28$). Once again, the significance and effect size of the clean results are small compared to the results from optogenetic mice. The results from these control experiments are displayed in Supplemental Figure 7. From these same control subjects, we examined the firing of narrow-spiking (NS) single units around laser onset. Out of the 13 NS units from the control group, 12 of them did not show a significant change in activity during the first 2 ms of laser onset. In contrast, when we looked at the 9 NS units from our experimental group, we found that 8 of the 9 units exhibited a statistically-significant decrease in spiking during the first 2 ms of laser onset. These results are shown in Supplemental Figure 5. The performance and spiking comparisons between conditions in our control group, along with the differences in narrow-spiking unit activity during laser onset between groups, strongly support the conclusion that the effects arise from optogenetic suppression, and not non-specific effects of illumination.
(17) Finally, the authors used a silicon probe with 100 um spacing. Although "single units" were defined based on	We have now added L-ratio and isolation distance as additional criteria for sorting single

their ISI violation, this criterion is not sufficient. Like many other published papers, an additional measurement, such as isolation distance, is required to quantify how well-separated each cluster is from the rest of clusters.

units. From the 'Spike extraction and clustering' section in Methods, lines 539-45:

To determine which of the remaining clusters were single-units (SU), we utilized the `sortingQuality` toolbox (<https://github.com/cortex-lab/sortingQuality>) to calculate isolation distances and L-ratios⁶⁵. SUs must 1) have less than 5% of inter-spike intervals below 2ms (Figure 1C), 2) an isolation distance above 15, and 3) an L-ratio below 0.25. These thresholds are consistent with values used in past studies on single-unit activity^{66, 67, 68}, and clusters that did not meet any of these criteria were deemed multi-units (MUs).

These additional criteria resulted in 82 out of our 124 identified units being deemed as SU, compared to the former value of 101. The most statistically-significant results of our main analysis remain the same, and the manuscript is now edited to reflect the new sample size and *p*-values.

Revised Figures

Figure 1

- In D, 50% is changed to ~50% to reflect the change in the manuscript from “50%” to “approximately 50%” in line 127, per reviewer comment #5.
- The data in E now reflects the new population of single units obtained after refined analysis.

Figure 2

F Maximal performance across all single units

- The titles of the example configurations shown in Ai, Bi, and Ci are changed from “Clean configuration, Masked configuration #1, and Masked Configuration #2” to “Target only (Clean) Hotspot”, “Target + Masker (Masked) Hotspot”, and “Target + Masker Coldspot”, respectively.
- Target and masker location axes in D, E, and F show direction from contralateral locations to ipsilateral locations.
- The data in F now reflect the single units from the refined analysis. F also has a new title “Maximal performance across all single units”.

Figure 3

- Aiii now has a title “Inset” to better reflect that data in Aiii is from Ai and Aii.
- B is now split into Bi (Control grid) and Bii (Laser grid).
- Bi and Bii now have dashed outlines on the masked hotspots mentioned in the manuscript (see lines 212-3).
- The data in C reflect the changes to the single-unit analysis.

Figure 4

- All data shown here reflect the refined single-unit analysis.
- A, B, and C titles now better reflect the features of responses that each performance metric is based on, instead of the distance themselves. The specific spike train distance measures are included in the y-axis label for each subplot.
 - ISI-distance performance -> Firing rate modulation-based performance
 - RI-SPIKE-distance performance -> Spike-timing-based performance
 - Spike count distance performance -> Mean firing rate-based performance

Figure 5

- All data shown here reflect the refined single-unit analysis.
- A and B now both have titles to better reflect what each measure calculates. A is titled “Dissimilarity of target responses” while B is titled “Response reproducibility”.

Figure 6

- All data shown here reflect the refined single-unit analysis.
- The y-axis of the histogram in 6A is relabeled “# hotspots” instead of “Count”.

Figure 7

- The y-label on the middle plot, a re-plotting of Figure 6A with a semi-logarithmic x-axis, is re-labeled “# hotspots” instead of “Count”.

Reviewers' comments:

Reviewer #1 (Remarks to the Author):

The authors substantially revised and improved the manuscript. They have included additional data and analysis, most importantly, they demonstrate the specific expression of Arch in PV interneurons and the rapid and selective inhibition of fast-spiking, putative PV interneurons in vivo during illumination. All concerns raised were satisfactorily addressed in the revised manuscript.

Reviewer #2 (Remarks to the Author):

The Authors partially addressed my points and the manuscript has significantly improved. However, some major methodological gaps remained.

Major points

1. The 'non-expressing' control is encouraging, but does not control for possible non-specific effects of virus expression. Therefore, YFP-control would be better, as suggested by my first point in the previous round. The laser-effect ('Control-Laser') should be statistically compared directly between Arch and non-expressing groups to derive statistically convincing results, especially since the average performance is nominally smaller during laser stimulation in the non-expressing animals as well. Is the 'control minus laser' difference significantly smaller in the non-expressing vs. the Arch expressing mice?

2. The electrode and optic fiber placement have not been demonstrated. I consider this necessary (as laid out in my previous review).

I believe that openness and transparency can increase the fairness of peer review. Therefore, I decided to sign my reviews.

Balazs Hangya

Reviewer #4 (Remarks to the Author):

In the revised manuscript, the authors have addressed all the technical issues I raised initially. I believe that the manuscript will be improved further by addressing the following issues:

MAJOR

1. Be consistent regarding reporting P-values. For example, lines 213-214. See also the minor comment below.

2. In Discussion, Figure 8 is too speculative. Unless it has been validated by a computational simulation, it should be provided as a supplementary figure.

3. Regarding subjects, clarify their genetic background. Are all genetic lines on C57BL/6J background, CBA/CaJ, or something else? This is very important for mouse auditory research.

MINOR

1. Line 115. It may be helpful to describe that the recording was done in an unanesthetized condition.

2. Line 170. I found "p <= 0.05" odd.

Reviewers' comments:

Reviewer #1 (Remarks to the Author):

The authors substantially revised and improved the manuscript. They have included additional data and analysis, most importantly, they demonstrate the specific expression of Arch in PV interneurons and the rapid and selective inhibition of fast-spiking, putative PV interneurons in vivo during illumination. All concerns raised were satisfactorily addressed in the revised manuscript.

Reviewer #2 (Remarks to the Author):

The Authors partially addressed my points, and the manuscript has significantly improved. However, some major methodological gaps remained.

Major points	
1. The 'non-expressing' control is encouraging but does not control for possible non-specific effects of virus expression. Therefore, YFP-control would be better, as suggested by my first point in the previous round. The laser-effect ('Control-Laser') should be statistically compared directly between Arch and non-expressing groups to derive statistically convincing results, especially since the average performance is nominally smaller during laser stimulation in the non-expressing animals as well. Is the 'control minus laser' difference significantly smaller in the non-expressing vs. the Arch expressing mice?	Regarding the first point, we would like to clarify that all our experiments were done in transgenic PV-Arch mice (PV-Cre crossed with Ai40). We did not use viruses (e.g., AAV, lentivirus) in this study. Thus, we believe that the use of PV-Cre control mice is the most rigorous control for the transgenic mouse experiment as evidenced in the literature by ourselves and others (Weible et al., 2017; Piscopo et al., 2018; Gritton et al., 2019). For the statistical analysis, we ran an un-paired, 2-sample t-test on the 'control minus laser' performance difference in each subject group. The t-test yielded a significant difference between groups in both clean ($p = 4.62e-05$, $d = -0.87$) and masked ($p = 0.0057$, $d = -0.78$) performances, which indicates that this performance decrease is significantly smaller in the control animals. We ran similar analyses on driven firing rate and found statistical significance in both clean ($p = 2.69e-09$, $d = 1.35$) and masked ($p = 0.0187$, $d = 0.66$) trials, which indicates that the increase in firing rate during laser trials is higher within Arch-expressing mice. These results are included in the legend of Supplemental Figure 8. Finally, we analyzed the 'control minus laser' difference in laser onset firing rate within NS cells in both Arch- and non-Arch-expressing subjects, as shown in Supplemental Figure 6. An un-paired 2-sample t-test yielded significance between the two groups ($p = 8.42e-05$, $d = 2.13$), which indicates that NS cells within Arch-expressing subjects exhibited a larger decrease in spiking during laser onset.
2. The electrode and optic fiber placement have not been demonstrated. I consider this necessary (as laid out in my previous review).	To address this point, we have now performed histological analysis on the control ($n = 5$) group. In all control subjects, histology confirms that recording electrodes were in A1 consistent with the electrophysiology results (see Figure S2 – CSD analysis and sink onset latencies of 17ms consistent with responses in A1 of mice (Seybold et al., 2012; Bhumika et al., 2020)). Supplemental Figure 1 shows a representative section from one of these controls. Optical fibers do not produce tract marks as they are lowered only to the cortical surface between the innermost shanks (lines 430, 477, 857). A surgery example is shown below. The vast majority of brain tissue from Arch mice ($n = 9$) was used to quantify Arch expression, as shown in Supplemental Figure 4.

Reviewer #4:

In the revised manuscript, the authors have addressed all the technical issues I raised initially. I believe that the manuscript will be improved further by addressing the following issues:

Major points	
1. Be consistent regarding reporting P-values. For example, lines 213-214. See also the minor comment below.	All p -values previously reported as " $p < 1e-04$ " are now reported with 3 significant figures and magnitude (e.g., $p = 2.02e-09$ in line 224).
2. In Discussion, Figure 8 is too speculative. Unless it has been validated by a computational simulation, it should be provided as a Supplemental figure.	Figure 8 has now been moved to Supplemental Information as Supplemental Figure 9.
3. Regarding subjects, clarify their genetic background. Are all genetic lines on C57BL/6J background, CBA/CaJ, or something else? This is very important for mouse auditory research.	The subjects' genetic lines (C57BL/6J background) are now given in the Methods section, line 422.
Minor points	
1. Line 115. It may be helpful to describe that the recording was done in an unanesthetized condition.	The unanesthetized condition is now described in lines 115: "different layers in ACx of unanesthetized PV-Arch transgenic mice", and 462: " Unanesthetized subjects were exposed to...".
2. Line 170. I found " $p \leq 0.05$ " odd.	Line 170 now reads " $p < 0.05$ ".

References

- Bhumika S, Nakamura M, Valerio P, Solyga M, Linden H, Barkat TR (2020) A Late Critical Period for Frequency Modulated Sweeps in the Mouse Auditory System. *Cereb Cortex* 30:2586-2599.
- Gritton HJ, Howe WM, Romano MF, DiFeliceantonio AG, Kramer MA, Saligrama V, Bucklin ME, Zemel D, Han X (2019) Unique contributions of parvalbumin and cholinergic interneurons in organizing striatal networks during movement. *Nat Neurosci* 22:586-597.
- Piscopo DM, Weible AP, Rothbart MK, Posner MI, Niell CM (2018) Changes in white matter in mice resulting from low-frequency brain stimulation. *Proc Natl Acad Sci U S A* 115:E6339-E6346.
- Seybold BA, Stanco A, Cho KK, Potter GB, Kim C, Sohal VS, Rubenstein JL, Schreiner CE (2012) Chronic reduction in inhibition reduces receptive field size in mouse auditory cortex. *Proc Natl Acad Sci U S A* 109:13829-13834.
- Weible AP, Piscopo DM, Rothbart MK, Posner MI, Niell CM (2017) Rhythmic brain stimulation reduces anxiety-related behavior in a mouse model based on meditation training. *Proc Natl Acad Sci U S A* 114:2532-2537.